# Cell type signatures in cell-free DNA fragmentation profiles reveal disease biology

Kate E. Stanley[1,2], Tatjana Jatsenko[1], Stefania Tuveri [1], Dhanya Sudhakaran[1], Lore Lannoo [3], Kristel Van Calsteren[3], Marie de Borre[4], Ilse Van Parijs[5], Leen Van Coillie[5], Kris Van Den Bogaert[5], Rodrigo De Almeida Toledo [6], Liesbeth Lenaerts [7], Sabine Tejpar [8], Kevin Punie [9], Laura Y. Rengifo [10], Peter Vandenberghe[10,11], Bernard Thienpont [4] & Joris Robert Vermeesch [1] ✉

Circulating cell-free DNA (cfDNA) fragments have characteristics that are specific to the cell types that release them. Current methods for cfDNA deconvolution typically use disease tailored marker selection in a limited number of bulk tissues or cell lines. Here, we utilize single cell transcriptome data as a comprehensive cellular reference set for disease-agnostic cfDNA cell-of-origin analysis. We correlate cfDNA-inferred nucleosome spacing with gene expression to rank the relative contribution of over 490 cell types to plasma cfDNA. In 744 healthy individuals and patients, we uncover cell type signatures in support of emerging disease paradigms in oncology and prenatal care. We train predictive models that can differentiate patients with colorectal cancer (84.7%), early-stage breast cancer (90.1%), multiple myeloma (AUC 95.0%), and preeclampsia (88.3%) from matched controls. Importantly, our approach performs well in ultra-low coverage cfDNA datasets and can be readily transferred to diverse clinical settings for the expansion of liquid biopsy.

Many cell types release cell-free DNA (cfDNA) into plasma, urine, and other extracellular fluids[1] upon apoptosis, necrosis, or active secretion. CfDNA fragments are derived from nuclease-processed nucleosome arrays which consist of 147 base pairs (bp) of DNA wrapped around a histone octamer core with approximately 20–40 bp of unbound linker DNA interspersed[2]. Nucleases preferentially cleave genomic DNA at accessible linker regions resulting in a large portion of mono-nucleosomal cfDNA fragments that span sites that are protected in vivo[3,4]. CfDNA fragmentation is therefore highly correlated with nucleosome organization, and can be used to infer nucleosome positioning[5,6], transcription factor binding[7], gene expression[8], and DNA methylation status[9] in the tissues-of-origin.

In plasma, cfDNA is largely hematopoietic in origin[10], but its composition alters under many (patho)physiological conditions. In pregnancy, 2–20% of cfDNA originates from the placenta, which allows for non-invasive prenatal screening (NIPS) of fetal aneuploidy[11]. In cancer, tumor-derived cfDNA is leveraged to screen, diagnose, and monitor the treatment of patients with various cancer types[12]. Altered cfDNA profiles have also been noted in transplant rejection[13], autoimmune disease[14], infection[15,16], myocardial infarction[17], and stroke[18], which may be the result of aberrant contributions from affected tissue types. To date, many strategies to trace cfDNA tissues-of-origin are limited by the need for genetic differences between the contributing tissue types (e.g., the fetal versus maternal genome, or mutated tumor

[1]Department of Human Genetics, Laboratory for Cytogenetics and Genome Research, KU Leuven, Leuven, Belgium. [2]Department of Biosciences and Nutrition, Karolinska Institute, Huddinge, Sweden. [3]Department of Gynecology and Obstetrics, University Hospitals Leuven, Leuven, Belgium. [4]Department of Human Genetics, Laboratory for Functional Epigenetics, KU Leuven, Leuven, Belgium. [5]Center for Human Genetics, University Hospitals Leuven, Leuven, Belgium. [6]Vall d'Hebron Institute of Oncology, Barcelona (VHIO), Spain. [7]Department of Oncology, Gynecological Oncology, KU Leuven, Leuven, Belgium. [8]Department of Oncology, Molecular Digestive Oncology, KU Leuven, Leuven, Belgium. [9]Multidisciplinary Breast Centre, Leuven Cancer Institute, University Hospitals Leuven, Leuven, Belgium. [10]Department of Human Genetics, Laboratory of Genetics of Malignant Diseases, KU Leuven, Leuven, Belgium. [11]Department of Hematology, University Hospitals Leuven, Leuven, Belgium. ✉e-mail: joris.vermeesch@kuleuven.be

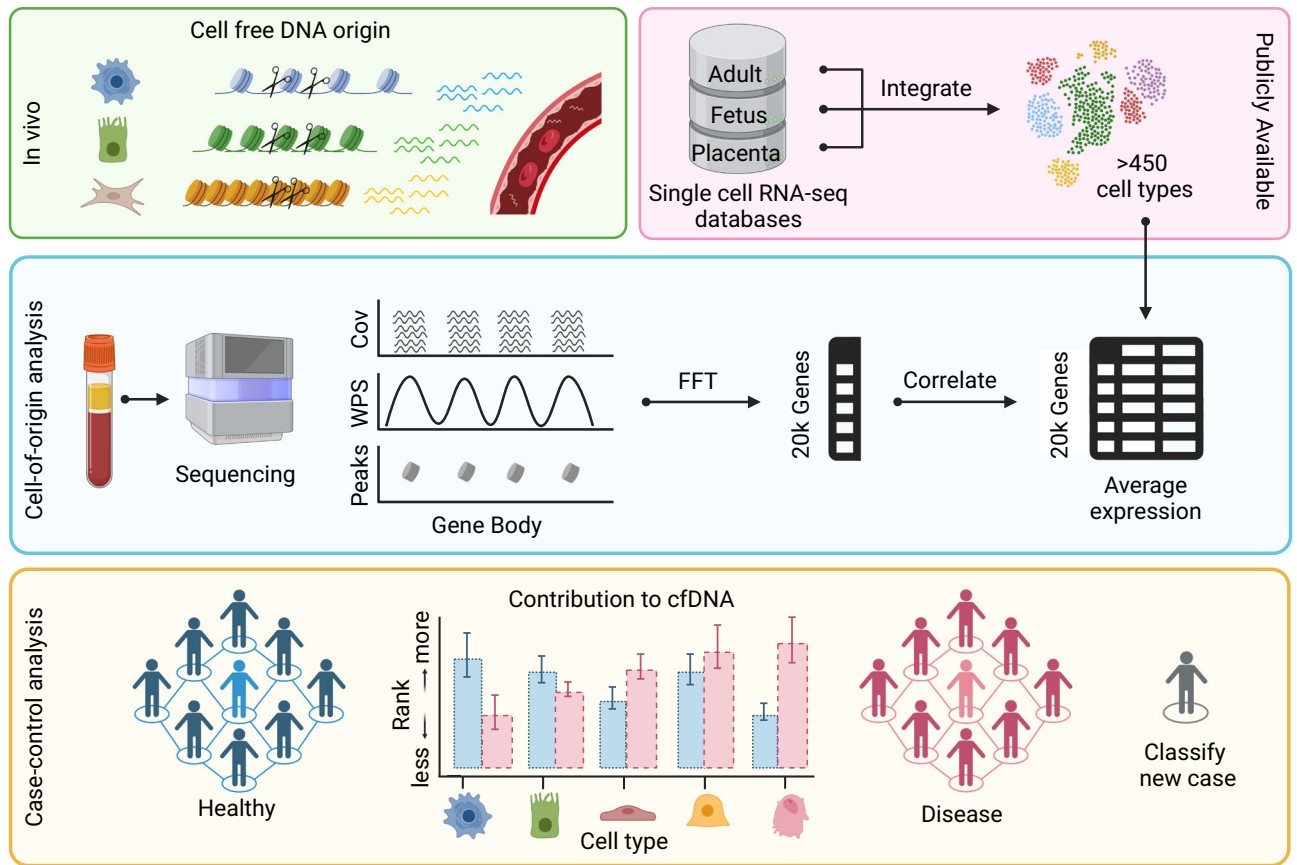

**Fig. 1 | Schematic of study approach.** Cell-free DNA (cfDNA) is fragmented and released by many different cell types into bodily fluids, including plasma. CfDNA fragment coverage (cov) reflects nucleosome positioning in the cell types of origin and can be quantified using a window protection score (WPS). We integrate publicly available single-cell RNA sequencing (seq) data from the Tabula Sapiens database (adult), Fetal Cell Atlas database (fetal), and Vento-Tormo et al. (2018) (placenta) to deconvolute over 450 potential cell type contributions to plasma cfDNA. Average gene expression per annotated cell type is correlated with fast Fourier transformed (FFT) cfDNA WPS in ~20 thousand (k) gene bodies. Cell types are then ranked based on the strength of this correlation. We compare cell type ranks between diverse cohorts of health and disease to identify cell types that are relatively more or less represented in plasma cfDNA. The disease status of new cases can be predicted by building classifiers that use cell type ranks as input.

DNA versus healthy tissue DNA). To overcome this limitation, non-genetic means of tracing the origins of cfDNA molecules have been explored as an alternative.

Non-genetic approaches are typically either methylation-based[19,20] or fragmentation-based[21]. Most methods require pre-selecting differentially methylated or accessible regions from epigenetic reference databases in specific tissues or diseases of interest. Although specific, these tailored approaches are not broadly applicable and not well-suited for the discovery of disease biology. Among the fragmentation-based approaches, Snyder et al. (2016) conducted a search for potential contributors to cfDNA in healthy and malignant states using bulk transcriptome data in 76 tissues and cell lines[5]. Our work builds on this approach but is substantially more complete.

The rapid advancement of single-cell technologies has provided a near-saturated reference set of cell type transcriptomes in the human body[22] and developing fetus[23]. We developed an approach that utilizes newly available single-cell transcriptome atlases to de novo scan cfDNA fragmentation profiles for a comprehensive set of potential cell type contributors in health and disease (Fig. 1). We demonstrate the broad applicability of this approach in multiple cancer types and pregnancy complications for disease biology exploration, biomarker discovery, and patient classification. Importantly, we show that cell type signatures can be inferred from ultra-low coverage cfDNA data, making the approach compatible with existing clinical datasets. We expect the approach will broaden the clinical utility of standard-of-care genome-wide NIPS and liquid biopsy data in various pathologies, including cancer.

## Results

### Global and local nucleosome dynamics are recapitulated across different cfDNA sequencing depths

We first tested if nucleosome positioning could be reliably captured using a sliding window protection score (WPS) in our genome-wide cfDNA paired-end sequencing data. The WPS provides a continuous score that quantifies the periodic nucleosome protection of DNA from nuclease digestion in vivo (Fig. 2A). As previously described[5], the per-base WPS is the number of fragments that span a 120 base pair (bp) window centered at a given genomic coordinate minus the number of fragments with an endpoint in the same window. We applied a peak calling algorithm (see "Methods" section) on the WPS curve to identify the position of individual nucleosomes in each sample.

In 230 healthy control cfDNA samples sequenced at 35-fold coverage ($n = 1$), 10-fold coverage ($n = 139$), or <0.3-fold coverage ($n = 90$), an average of approximately 10 M, 3 M, and 5 K nucleosome peaks were called, respectively. The position of nucleosome calls was concordant with the location of nucleosomes mapped by Snyder et al. (2016) using the same method in a sample sequenced at 231-fold coverage ("CH01", 12.9 million peak calls) (Fig. 2B) and nucleosomes mapped using an in silico method by Kaplan et al. (2009) (Supplementary Fig. 1). In each sample, the mode distance between adjacent

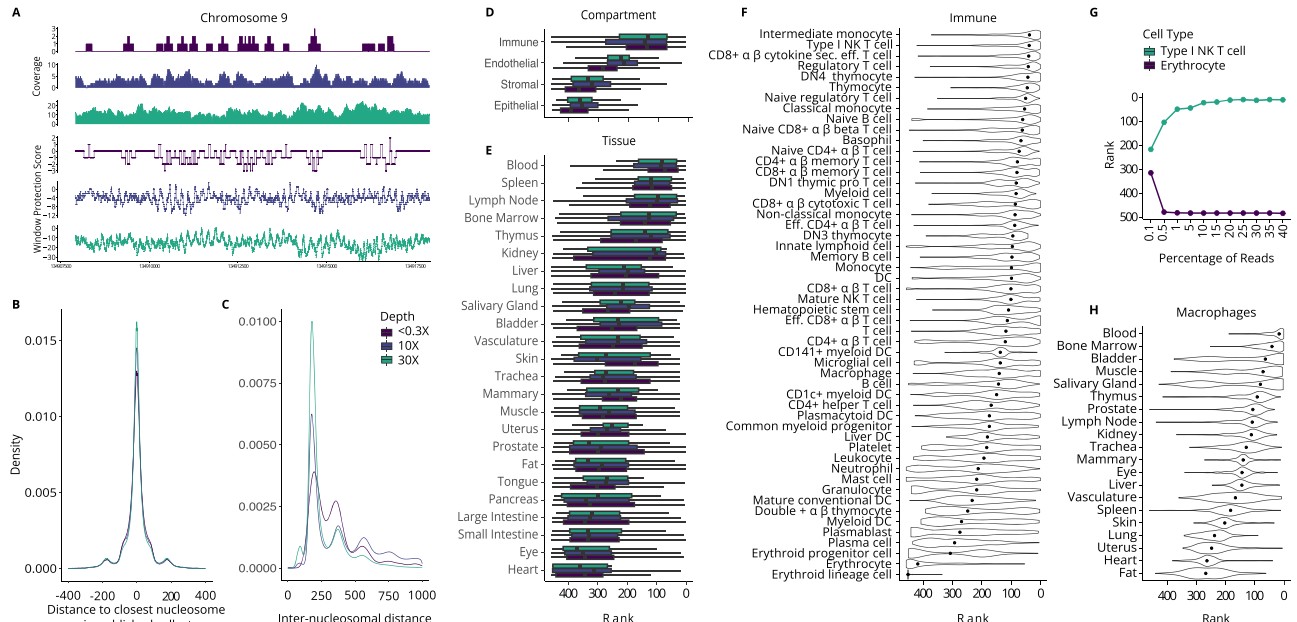

**Fig. 2 | Cell type signatures inferred from cell-free DNA fragmentation are preserved at ultra-low sequencing depths. A** Tracks for sequencing coverage and window protection scores (WPS) are shown for sample "GC01" sequenced at 35-fold (35×) coverage and a randomly selected sample sequenced at 10-fold (10×) and <0.3-fold (<0.3×) coverage for the first 5 kilobases (kb) of the *FCN1* gene. Per-base WPS was calculated by taking the number of fragments that span a 120 base pair (bp) window minus the number of fragments that have an endpoint within that same window centered at a given genomic coordinate. **B** The distance (bp) between the genomic coordinate of each WPS peak call in the set of healthy control samples and the closest WPS peak call in published sample "CH01" from Snyder et al. (2016) is plotted. Distances are grouped by sequencing coverage of the healthy control samples: 35-fold ($n=1$), 10-fold ($n=139$), and <0.3-fold ($n=90$) coverage. The published call set in sample "CH01" was sequenced at 231-fold coverage (12.9 million peaks). **C** The distance (bp) between adjacent peak calls (i.e., inter-nucleosomal distance) is plotted genome-wide for all healthy control samples grouped by

sequencing coverage. Ranked correlation values between fast Fourier transformed (FFT) WPS in the first 10 kb of 19,536 genes and the average expression of these genes in 456 cell types from the Tabula Sapiens database are grouped by **D** compartment and **E** tissue for samples sequenced at 35-fold ($n=1$), 10-fold ($n=139$), and <0.3-fold ($n=90$) coverage. Ranked correlation values are stratified by cell type for the immune compartment **F** for all healthy control samples ($n=230$). Boxplots indicate median, interquartile range (IQR), and whiskers for 1.5× IQR. Cell types were annotated by the original publication with four compartments (i.e., immune, epithelial, endothelial, or stromal) and originated from 24 different tissue types biopsied by the Tabula Sapiens consortium **G** Ranks for the highest (type I NK T cell) and lowest (erythrocyte) correlations are plotted for sample "GC01" when down-sampled **H** The macrophage rank distribution grouped by tissue-residency is plotted for all healthy control samples ($n=230$). NK natural killer, Eff. effector, DC dendritic cell, β beta, α alpha. Source data are provided as a Source Data file.

nucleosome calls was 187 bp reflecting the known nucleosome repeat length in mammals (Fig. 2C). At regulatory regions where nucleosomes have more defined positions (e.g., promotors), we were also able to recapitulate known dynamics across sequencing depths (Supplementary Fig. 2)[24]. Together, these findings indicate that mono-nucleosomal signals are preserved in cfDNA fragmentation profiles even at ultra-low sequencing coverage.

### Cell type signatures inferred from cfDNA fragmentation profiles are mainly hematological

We then asked if the WPS could be used to infer the cell type composition of cfDNA in plasma from our healthy individuals. To do this, we applied a fast Fourier transformation (FFT) on WPS curves in the first 10 kilobase (kb) of 19,536 Ensembl genes and took the mean FFT intensity at the 196–199 bp range (see "Methods" section). This range corresponds to a wider inter-nucleosomal distance observed in the body of weakly transcribed genes. Intensities at this wavelength should therefore be negatively correlated with gene expression in cell types that contribute to the cfDNA population. We correlated FFT-WPS with average gene expression in 456 cell types from the Tabula Sapiens database and ranked their relative contribution to cfDNA based on the strength of correlation.

The strongest correlations for all cfDNA samples across all sequencing depths were with immune cell types followed by endothelial, stromal, and epithelial cell types as expected in plasma from healthy individuals (Fig. 2D). Accordingly, when grouping by tissue,

those that primarily produce and circulate immune cells, including the blood, thymus, lymph node, bone marrow, and spleen, were ranked more highly than other primary tissues (Fig. 2E). We next stratified the immune signal by cell type. In line with previous reports, monocytes and lymphocytes contributed the most to cfDNA in the healthy state[25,26]. In particular, the strongest correlations were made with classical and intermediate monocytes, regulatory T cells, natural killer T cells, CD8+ T cells, and naïve B cells in the healthy state compared to less common immune cell types such as plasma cells, platelets, and dendritic cells (Fig. 2F). As a negative control, we observed the weakest correlation for all samples with erythrocytes which do not contain DNA and are therefore not expected to contribute to cfDNA. The highest and lowest correlations in the 35-fold coverage sample ("GC01") are robust to downsampling to 0.5% of sequencing reads (~4 M or 0.1-fold) (Fig. 2G, Supplementary Figs. 3, 4).

The Tabula Sapiens database provides distinct transcriptional profiles for the same immune cell type shared across multiple tissues. When stratifying the same immune cell type by tissue residency, we observed variable contributions to cfDNA (Supplementary Fig. 5)[22]. This was particularly evident for tissue-resident macrophages which are well known to differentiate to fulfill niche-specific functions (Fig. 2H)[22,27]. Macrophages of the blood and bone marrow contributed most to cfDNA in plasma consistent with our expectation, followed by macrophages in other primary tissues. Importantly, we also compared immune cells from sex-specific organs between males and females. Interestingly, immune cells that were ranked significantly different

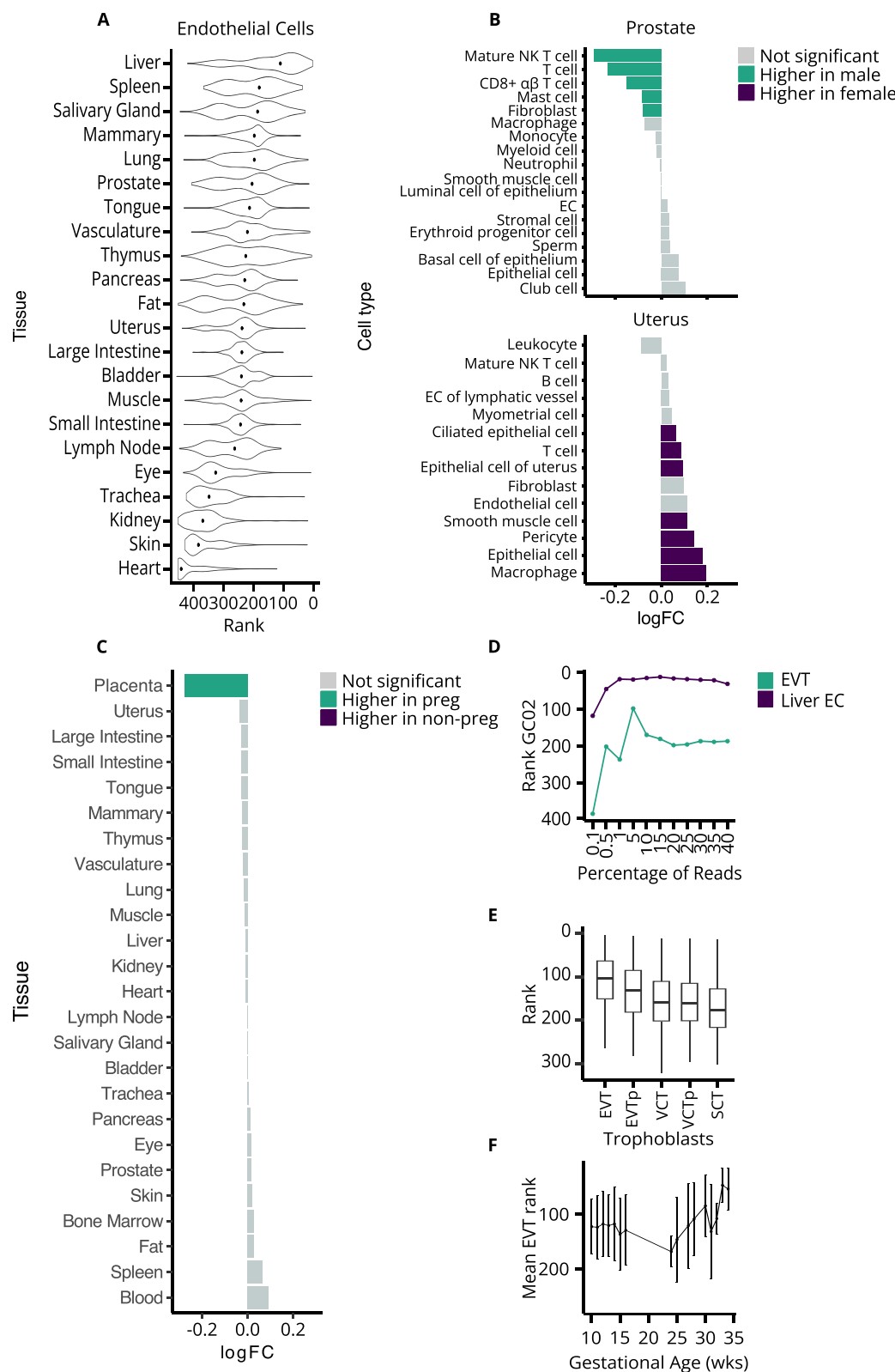

between males and females, were ranked higher in females when they were uterus-derived, and higher in males when they were prostate-derived (Fig. 3B). This substantiates our ability to stratify signals from immune cells based on their tissue residency.

**Sex-specific cell types contribute to cfDNA in the healthy state**
Non-hematopoietic contributions to cfDNA are also expected in healthy individuals[25,26]. When stratifying endothelial cells by tissue, a

striking correlation was observed for those of the liver compared to other tissues as supported by the literature[25] (Fig. 3A; Supplementary Fig. 6). A strong correlation with endothelial cells of the liver was robust to downsampling of a 35-fold coverage control sample ("GC02") to 0.5% of reads (~4 M or 0.1X) (Fig. 3D). To test if our method was sensitive to non-hematopoietic contributions from other tissues, we again compared cell type contributions to cfDNA from sex-specific tissues between males and females (Fig. 3B). Among non-

**Fig. 3 | Sex-specific and pregnancy-specific cell type contributions to cfDNA.**
**A** Endothelial cell rank distribution grouped by tissue is plotted for healthy non-pregnant control samples. **B** Cell type rank log$_2$ fold change (FC) between healthy males and healthy non-pregnant females is visualized for sex-specific tissues (i.e., prostate and uterus). Cell types were either significantly higher in males (green), significantly higher in females (purple) or not significant (gray). *P*-values were generated using a two-sided Wilcoxon rank-sum test and considered significant if the corrected Benjamini & Hochberg (BH) *p*-value ≤ 0.05. **C** Rank log$_2$ FC between healthy non-pregnant and healthy pregnant individuals is plotted for each tissue. Cell types per tissue were either significantly higher in pregnant individuals (green), significantly higher in non-pregnant individuals (purple) or exhibited no significant

difference (gray). **D** Rank correlations for healthy pregnant control sample "GC02" are shown for liver endothelial cells and extravillous cytotrophoblasts (EVTs) when down-sampled from 35-fold sequence coverage. **E** Trophoblast cell rank distribution grouped by sub-type is plotted for healthy pregnant control samples (*n* = 331). Boxplots indicate median, interquartile range (IQR), and whiskers for 1.5× IQR.
**F** The mean EVT rank with a standard deviation error bar is shown across healthy pregnant controls (*n* = 331) sampled at different gestational ages (wks weeks), VCT villous cytotrophoblast, SCT syncytiotrophoblast, p proliferative, preg pregnant, NK natural killer, EC endothelial cell, β beta, α alpha. Source data are provided as a Source Data file.

hematopoietic cell types of the uterus, all differentially ranked cell types were ranked higher in females than males. This includes pericytes, vascular smooth muscle cells, (ciliated) epithelial cells, and endothelial cells (of the lymphatic vessel). We found no significant difference for the remaining stromal cell types of the uterus (i.e., myometrial, fibroblast) between females and males, which may be because they have a very low or absent contribution to cfDNA. Conversely, all differentially ranked cell types of the prostate were ranked higher in males than females (Fig. 3B). Together, these findings support our ability to identify non-hematopoietic cell type contributions in cfDNA data from tissues that are expected to have a minor contribution.

### Pregnancy-specific cell types contribute to cfDNA in the healthy state

As a final validation, we tested for an expected placental contribution to cfDNA during pregnancy. We obtained retrospective ultra-low (0.1–0.3-fold) coverage cfDNA sequencing data in 301 healthy first-trimester pregnancies that underwent routine non-invasive prenatal screening (NIPS) and also prospectively sampled 30 healthy pregnancies later in gestation (24–34 weeks). Unfortunately, the Tabula Sapiens database does not include placental tissue, so we added publicly available single-cell RNA sequencing data from first-trimester placenta to our reference[28]. We compared cell type ranks in healthy pregnancies to cell type ranks in the set of low-coverage non-pregnant healthy individuals (*n* = 90). When grouping by tissue, the only differentially ranked cell types between pregnant and non-pregnant individuals were from the placenta (Fig. 3C).

When stratifying trophoblast cell types of the placenta among healthy pregnant individuals, extravillous trophoblasts (EVT) were the most highly ranked, followed by proliferative EVT, villous cytotrophoblasts (VCT), proliferative VCT, and syncytiotrophoblasts (SCT) (Kruskal–Wallis, *P* = 5 × 10$^{-86}$; Fig. 3E). This progression mirrors the trophoblast differentiation trajectory as cells detach from the villous tree and invade the maternal decidua where fetal cfDNA can enter the maternal circulation[28]. A strong correlation with EVTs was robust to downsampling of the 35-fold coverage pregnant control sample ("GC02") to 0.5% of reads (~4 M or ~0.1-fold) (Fig. 3D). When comparing across healthy pregnancies sampled between 10 and 34 weeks' gestation, the rank of trophoblast cells increased with the gestational age of our samples after 28 weeks which is in line with previous genetic-based estimates of the fetal cfDNA fraction across gestation[29] (Kruskal–Wallis, *P* = 0.023; Fig. 3F).

### Cell types aberrantly contribute to cfDNA in cancer patients

We next asked if we could detect aberrant cell type contributions in plasma cfDNA from cancer patients. We previously demonstrated that large-scale cfDNA coverage patterns vary in different cancers compared to the healthy state[30], and we reasoned that those differences may be driven by the contribution of affected cell types. To test this, we ranked cell types in cfDNA data from individuals with colorectal cancer (*n* = 16) and breast cancer (*n* = 52) sequenced at 10-fold coverage and compared the rankings to those generated in healthy

individuals sequenced at similar coverage and matched for sex (Supplementary Table 1).

In the colorectal cancer cohort (*n* = 16, 63% ≥ stage III), we identified 122 overrepresented cell types and 134 underrepresented cell types compared to 139 healthy individuals (Fig. 4A, Supplementary Data 1). The majority of cell types with an increased rank were epithelial cells (*n* = 48; 40%), followed by stromal (*n* = 28; 23%), endothelial (*n* = 24; 20%), and immune cells (18%). Intestinal cells accounted for 33% of the significantly up ranked epithelial cells and included Paneth cells, goblet cells, enterocytes, enteroendocrine cells, transit amplifying cells, duodenum glandular cells, tuft cells, and intestinal crypt stem cells. The most overrepresented cell type in colorectal cancer patients compared to healthy individuals was intestinal CD4+ α/β T cells (*p* < 10$^{-6}$, fold change = 2.1). Moreover, the rank of intestinal CD4+ α/β T cells was positively correlated (Pearson's *r* = 0.72, *P* = 0.0016) with an independent measure of tumor fraction in cfDNA (ichorCNA[31]) (Fig. 4B). The vast majority (93.3%) of the underrepresented cell types in the colorectal cancer cohort versus controls were immune cell types that highly contribute to cfDNA in the healthy state, but that are down ranked in colorectal cancer patients by the relative increase in intestinal T-lymphocytes and epithelial cells which do not typically contribute.

In the breast cancer cohort (*n* = 52), we detected 73 overrepresented and 83 underrepresented cell types compared to 88 healthy female individuals (Fig. 4C, Supplementary Data 2). Unlike the colorectal cancer cohort, we did not identify any epithelial cells from the primary affected tissue (i.e., mammary) among the overrepresented cell types. This may be due to the overrepresentation of cases with early stage breast cancer in our cohort (58% stage I and 19% stage II), which is associated with a low fraction of circulating tumor-derived cfDNA. Among the cell types ranked higher in breast cancer, 80% were immune cells, with macrophages showing a particularly high relative ranking. Macrophages are thought to facilitate early dissemination of cancer cells in pre-malignant breast cancer[32], which may explain their overrepresentation in our early-stage breast cancer cohort. We also observed a significantly reduced contribution of ciliated cells in the breast cancer cohort compared to healthy individuals. The rank of macrophages and ciliated cells were not correlated with the ichorCNV tumor fractions we calculated, but as our cohort does not span a range of cancer stages, we have a limited ability to correlate cell types with disease progression (Fig. 4D).

We next tested if our method could detect cell type-specific disease signatures in ultra-low-coverage data by analyzing cfDNA samples from a cohort of multiple myeloma patients sequenced at <0.3-fold coverage (25% stage I, 54% stage II, and 21% stage III). FFT-WPS was generated gene-by-gene in the same way as the high-coverage samples and correlated with gene expression in the same 456 cell types. We compared the cell type ranks in the ultra-low-coverage multiple myeloma samples (*n* = 24) to cell type ranks generated in the set of ultra-low-coverage healthy control samples matched for sex (*n* = 90; Fig. 4G, Supplementary Table 1 and Supplementary Data 3). Multiple myeloma is a hematological cancer characterized by the infiltration of plasma cells in the bone marrow. In our analysis, plasmacytoid dendritic cells,

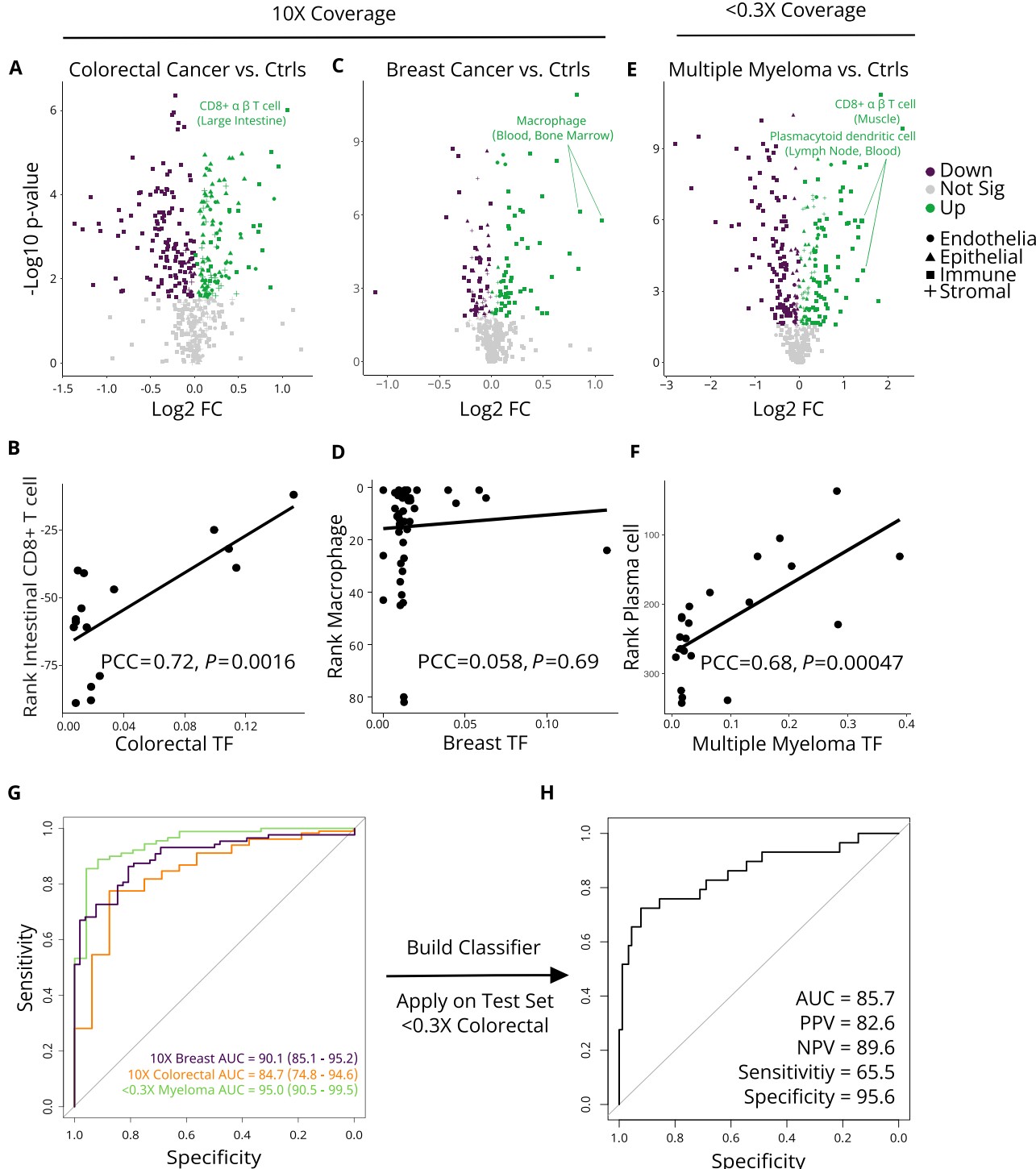

**Fig. 4 | Aberrant cell type contributions to plasma cell-free DNA in cancer.** Cell types with differential ranking between cancer cases and controls (ctrls) are visualized using a Volcano plot for **A** colorectal cancer **C** breast cancer, and **E** multiple myeloma. *P*-values were generated using a two-sided Wilcoxon rank-sum test and considered significant if the corrected Benjamini & Hochberg (BH) *p*-value ≤ 0.05. Fold change (FC) was calculated for each cell type by taking the ratio of the change between the mean rank of cases and the mean rank of controls over the mean rank of cases. The green, purple, and gray dots indicate that cell types were significantly higher ranked (Up), significantly lower ranked (Down) or did not exhibit significant differential ranking (Not Sig) in cases compared to controls, respectively. Labels for the top differentially ranked cell type(s) based on FC are provided. The ranks of the top differentially ranked cell type against an independent published measure of tumor fraction (TF) called ichorCNA are shown for

**B** colorectal cancer, **D** breast cancer, and **F** multiple myeloma. Lines represent the linear regression line. *P*-values and Pearson correlation coefficient (PCC) by a two-sided Pearson's product-moment correlation test. IchorCNA uses large-scale copy number variation in cfDNA sequencing data to estimate TF **G** Receiver operator characteristic (ROC) curves for support vector machine classification of cancer vs. healthy controls with leave-one-out cross-validation. Area under the ROC curve (AUC) is noted for each cancer type. The 95% confidence intervals of the ROC curves obtained from 2000 bootstrap iterations are shown **H** Performance of the colorectal model trained on internal 10-fold (10×) coverage cases is shown when applied on an external test set of colorectal cases sequenced at <0.3-fold (<0.3×) coverage. The ROC curve is plotted and the positive predictive value (PPV), negative predictive value (NPV), sensitivity, and specificity are noted. β beta, α alpha. Source data are provided as a Source Data file.

plasma cells, plasmablasts, B cells, and myeloid progenitor cells were among the top 20 up-ranked cell types in cases compared to controls. Plasma cell rank correlated strongly with ichorCNA tumor fraction in multiple myeloma cases (Pearson's $r = 0.68$, $P = 0.00047$; Fig. 4H) supporting the sensitivity of our approach at ultra-low sequencing depths.

## Cell type signatures predict cancer

Whereas the above case-control analyses identify disease signatures at the cohort level and can inform disease biology, we also asked if individual cancer patients could be distinguished from controls in a predictive analysis. Given that the sample sizes are limited compared to the number of features, we first applied unsupervised methods to visualize the data structure (see "Methods" section). Using Walktrap community detection we observed an uneven distribution of cases and controls in the clusters for all cancer types (Supplementary Fig. 7). These results were supported by an apparent separation between cases and controls in the t-distributed Stochastic Neighbor Embedding visualization (Supplementary Fig. 7).

We then trained a support vector machine with default hyperparameters and assessed performance of the supervised model using leave-one-out cross-validation for each cancer type. Receiver operating characteristic (ROC) analysis yielded an average area under the ROC curve (AUC) of 84.7% for colorectal cancer, 90.1% for breast cancer, and 95.0% for multiple myeloma (Fig. 4I). We then sought to validate our findings in an independent cohort. We obtained an additional set of 29 colorectal cancer cases sequenced at ultra-low (<0.3-fold) coverage at an external site. We applied the colorectal model trained on the internal 10-fold coverage data on the external <0.3-fold coverage data and achieved a positive predictive value of 82.6 and a negative predictive value of 89.6 (Fig. 4H). This demonstrated the ability of our model to generalize across sequencing depths and research centers for the classification of colorectal cancer.

We tested the added value of using single-cell transcriptomic references compared to bulk tissue transcriptomic references for cancer classification. For this analysis, we ranked the relative contribution of 50 tissues using bulk transcriptomic data downloaded from the Human Protein Atlas and GTEx (see "Methods" section). These tissues largely correspond to the tissues from the Tabula Sapiens database for which we have single-cell data. In our cohorts, using cell type features compared to using bulk tissue features boosted classification performance for all cancer types; the AUROC decreased from 95.0% to 87.1%, 84.7% to 73.4%, and 90.1% to 84.6% for multiple myeloma, colorectal cancer, and breast cancer, respectively (Supplementary Fig. 8).

We then benchmarked our method against two additional low-pass fragmentation-based cfDNA methods for cancer detection. IchorCNA uses binned genome-wide cfDNA fragment coverage to estimate tumor fraction in cancers with copy number alterations. In our cohorts, using ichorCNA tumor fractions as a prediction value resulted in an AUROC of 92.6 for multiple myeloma, 65.7 for breast cancer, and 64.2 for colorectal cancer, highlighting the major advantage of our method in patients with low tumor burden or copy number neutral cancer (Supplementary Fig. 9, Supplementary Data 4). We then benchmarked against Griffin as it does not rely on copy number alterations and has been demonstrated to perform well in early-stage cancers sequenced at low-coverage[33]. Griffin uses normalized fragment coverage at the binding sites of 270 transcription factors to capture changes in gene regulation for cancer prediction. Griffin performed better than our cell type features in the context of high-coverage later-stage colorectal cancer (AUROC 90.9% compared to 84.7%) but worse for high-coverage early-stage breast cancer (AUROC 77.4% compared to 90.2%) and for low-coverage later-stage multiple myeloma (AUROC 75.6% compared to 95.0%; Supplementary Fig. 10).

We then tested the classification performance of our cell type signatures in a multi-class setting using a one-vs-all support vector machine with default hyperparameters (see "Methods" section). We limited this analysis to breast and colorectal cancer as these samples were sequenced at the same depth along with matched controls. The one-vs-all SVM was able to correctly classify the majority of both breast (44/52; 84.6%) and colorectal (6/16; 37.5%) cancer patients as their respective classes with an overall accuracy of 71.0% (Supplementary Data 5). These results suggest that there are enough distinct cell type associations across cancer types for multi-class classification with reasonable performance.

## Cell type signatures are associated with pregnancy complications

We then tested for aberrant cell type contributions to cfDNA in pregnancy complications compared to the healthy pregnant state. We obtained first-trimester NIPS data in pregnancies with vanishing twin syndrome ($n = 102$) and pregnancies that went on to have a miscarriage ($n = 44$; Supplementary Data 6). We prospectively sampled 18 pregnancies with preeclampsia at diagnosis (24–34 weeks' gestation) and also prospectively sampled 30 healthy control pregnancies that were matched to our preeclampsia cases for gestational age at sampling, maternal age at conception, body mass index (BMI), and ethnicity (Supplementary Data 7, Supplementary Fig. 11). We expanded the number of cell types we searched for by adding gene expression data from the Fetal Cell Atlas which includes cell types from 15 different fetal tissues[23]. The ranks of 496 adult and fetal cell types were then compared between each pregnancy complication cohort and gestational age-matched healthy singleton pregnancies.

Preeclampsia is a multisystem pregnancy complication characterized by severe hypertension, affecting 2–8% of pregnancies worldwide[34]. Its etiology is poorly understood but thought to involve deficient placentation. Increased placental shedding has been previously reported in second-trimester preeclamptic pregnancies using genetic-based methods[35,36]. In 18 preeclamptic pregnancies sampled at the time of diagnosis, we identified AFP+ ALB+ cytotrophoblasts, neutrophils of the liver, and monocytes among the top 10 overrepresented cell types in cfDNA compared to matched control pregnancies (Fig. 5A, Supplementary Data 8). Liver damage is one of the most common sequelae in preeclampsia and was reported in 39% of our cases at diagnosis. The overrepresentation of liver-resident neutrophils may be a consequence of this liver damage. We also observed an underrepresentation of CD8+ α/β T cells, CD4+ α/β T cells, and naïve regulatory T cells in cfDNA from preeclamptic pregnancies compared to gestational age-matched control pregnancies. Using a support vector machine with leave-one-out cross-validation, we distinguished preeclamptic pregnancies at diagnosis from gestational age-matched control pregnancies with an AUC of 88.3% (Fig. 5B).

Finally, we investigated vanishing twin syndrome, where one fetus dies *in utero* and is absorbed partially or completely by the maternal circulation. Vanishing twin pregnancies account for approximately 30% of false positive NIPS results but often go unnoticed on ultrasound scans[37,38], underscoring the need to identify them for improved NIPS interpretation and clinical management. We compared 102 pregnancies with vanishing twin syndrome to 301 healthy singleton pregnancies sampled in the first trimester. Our hypothesis was that in pregnancies with a vanishing twin, fetal DNA fragments from the placenta as well as other fetal tissues might be absorbed and present in the maternal circulation. Interestingly, we observed stronger correlations with fetal endothelial cells in pregnancies with a vanishing twin compared to those without (Fig. 5C, Supplementary Data 9). This links cfDNA fragments in the maternal circulation to fetal tissues other than the placenta. We expected this finding to be specific to vanishing twin syndrome, and not more generally associated with fetal demise when

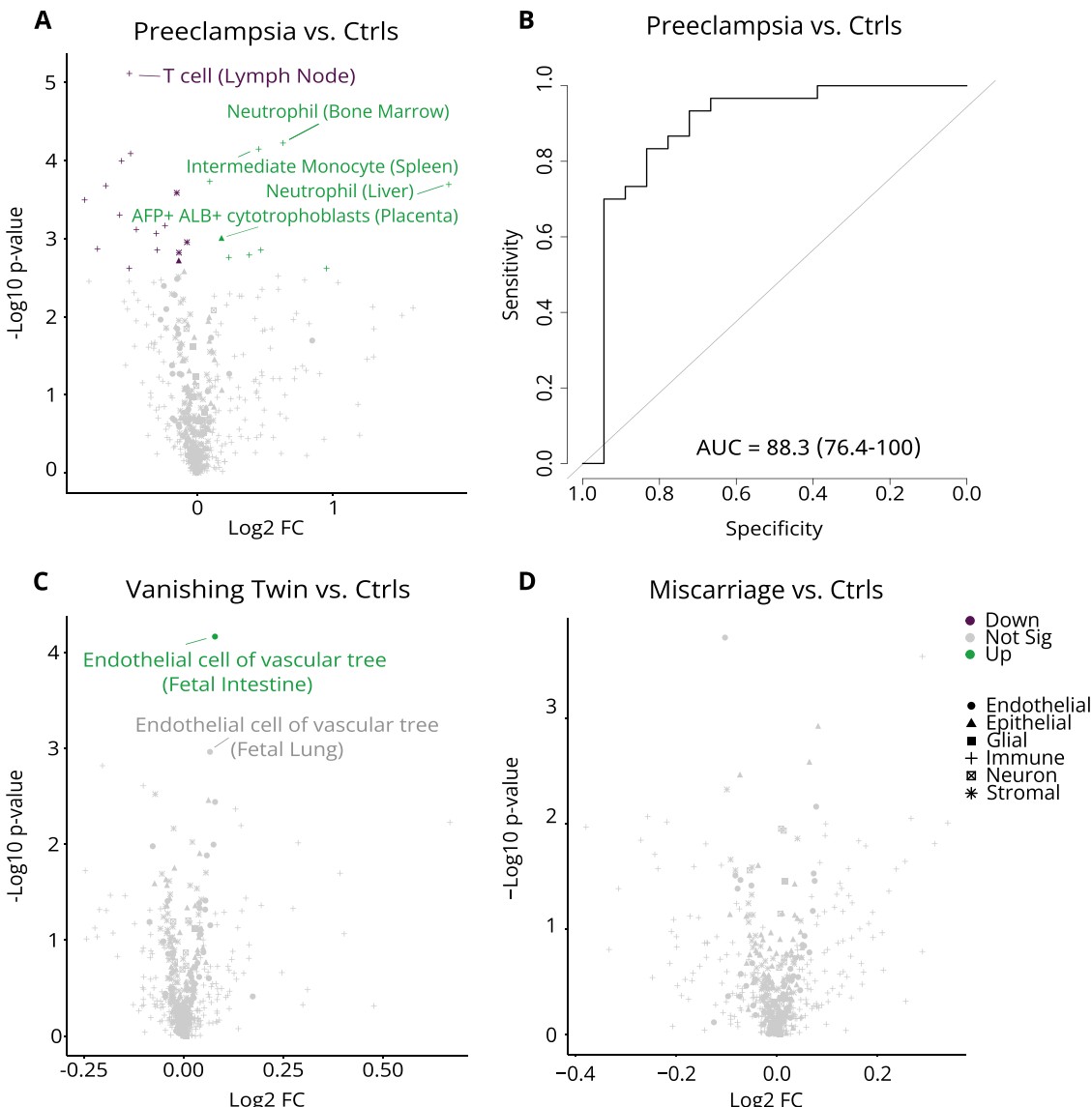

**Fig. 5 | Aberrant cell type contributions to plasma cell-free DNA in pregnancy complications.** Cell types with differential ranking between pregnancy complication cases and healthy pregnant controls (ctrls) are visualized using a Volcano plot for **A** preeclampsia **C** vanishing twin syndrome, and **D** miscarriage. *P*-values were generated using a two-sided Wilcoxon rank-sum test and considered significant if the corrected Benjamini & Hochberg (BH) *p*-value ≤ 0.05. Fold change (FC) was calculated for each cell type by taking the ratio of the change between the mean rank of cases and the mean rank of controls over the mean rank of cases. The green, purple, and gray dots indicate that cell types were significantly higher ranked (Up), significantly lower ranked (Down) or did not exhibit significant differential ranking (Not Sig) in cases compared to controls, respectively **B** Receiver operator characteristic (ROC) curve for support vector machine classification of preeclampsia vs. healthy pregnant controls with leave-one-out cross-validation is shown. Principal component analysis was used in the preeclampsia classification task. Area under the ROC curve (AUC) and the 95% confidence intervals of the ROC curve obtained from 2000 bootstrap iterations are shown. β beta, α alpha, AFP alpha-fetoprotein, ALB albumin. Source data are provided as a Source Data file.

the fetus is expelled from the uterus (i.e., miscarriage) and not absorbed. Consistent with our expectation, we did not detect signals from non-placental fetal cell types in 44 pregnancies that experienced a miscarriage shortly (<10 weeks) after cfDNA sampling compared to controls (Fig. 5D, Supplementary Data 6). In fact, there were no differentially ranked cell types in first-trimester pregnancies that went on to have a miscarriage compared to healthy control pregnancies (Supplementary Data 10). This is not surprising considering the diverse causes of fetal demise.

Although encouraging at the cohort level, our predictive analyses were not able to distinguish between individual pregnancies with a vanishing twin from controls with high performance (AUC < 60%). Given that a sub-group of cases appeared to be driving the observed disease signatures, a clear margin could not be drawn between the

cases and controls during the classification task. This is likely related to the time between fetal demise and blood sampling as well as the degree of absorption by the mother (i.e., partial or complete).

## Discussion

We integrate single-cell transcriptome atlases with cfDNA fragmentation patterns to perform a comprehensive characterization of plasma cfDNA cell type contributors. Our work builds on previous efforts to use bulk transcriptome data for cfDNA tissue-of-origin analysis[5] but is substantially more complete. Furthermore, current single-cell transcriptome databases provide more exhaustive cellular reference maps than single-cell methylation and chromatin databases used in other cfDNA deconvolution approaches[26,39,40]. In robust case-control analyses, we perform an unbiased search for aberrant cell type

contributions in multiple cancer types and pregnancy complications, providing evidence for established and emerging disease paradigms in oncology and prenatal care. One of the major benefits of capturing cell type signatures in cfDNA data is that the features are highly interpretable for clinical specialists and can be used to explore specific cell types of interest, more so than other less biologically interpretable fragmentation metrics.

Our validation tests in the healthy state reveal sex-specific and pregnancy-specific cell type contributions that substantiate our ability to identify both hematological and non-hematological signatures in cfDNA fragmentation profiles. Consistent with existing literature, we observe the greatest contribution to cfDNA from monocytes and lymphocytes and an absence of erythrocytes. Compared to fully differentiated erythrocytes, erythroid progenitor cells show a relatively stronger signal but appear to contribute to a lesser extent than was previously suggested[26,41]. In the healthy state, we observe a strong correlation with (DN4) thymocytes which supports recent reports of an enrichment of bone marrow-derived fragments in circulating nucleic acid populations[42]. Neutrophils were not among the top correlations despite accounting for a large portion of whole blood cells. We suspect that NETosis, a neutrophil-specific form of cell death that involves the fusion of chromatin with antimicrobial proteins to form extracellular mesh-like structures, may interfere with our ability to quantify neutrophil-derived fragments. We find it interesting that B lymphocytes appear among the top-ranked cell types along with T lymphocytes despite there being a 4:1 ratio of B to T cells expected in circulation[42]. This corroborates recent reports of a large discrepancy between cell-derived cell-free DNA and the counts of these cells in circulation[43,44].

In oncology, our findings strengthen reports of an abundance of intestinal CD4+ T cells in colorectal cancer patients[45] and a critical role for macrophages in the early dissemination of cancer cells in pre-malignant breast cancer[32]. Additionally, we identified a loss of cilia in early-stage breast cancer patients, supporting a recently proposed mechanism for the promotion of tumor growth[46,47]. In multiple myeloma patients, primary affected cell types, including plasma cells, plasmablasts, and plasmacytoid dendritic cells were among the top associations. These associations correlated with tumor fraction for colorectal cancer and multiple myeloma, but not early-stage breast cancer given the low tumor burden across samples. Liquid biopsy has revolutionized clinical oncology and is pushing past diagnosis to screening, prognosis, and personalized therapy. Our method allows for the future investigation of cell type-specific signals in pre-malignant populations, and across disease progression and treatment regimens for cancer cohorts with repeat sampling.

In the prenatal setting, we demonstrate that cfDNA-inferred cell type signatures can provide insights into disease biology. In pre-eclamptic pregnancies we identify placental signal coming specifically from AFP+ ALB+ cytotrophoblasts which have been previously detected in early but not full-term placentas[48]. The reduction of maternal serum alpha-fetoprotein (AFP) in the second trimester has been associated with a reduced risk of preeclampsia, preterm birth, and small for gestational age infants[49]. The presence of AFP and its receptor in the placenta has been associated with the transport of AFP between fetal and maternal circulations[48,49]. The observed persistence of these immature AFP+ cytotrophoblast cells in preeclamptic pregnancies may explain the reported changes in AFP concentration in the maternal blood. In addition, several studies suggest that AFP reduces proliferation of T-lymphocytes[50] which are underrepresented in our analysis in preeclampsia pregnancies compared to control pregnancies. The clinical utility of this approach for the pre-symptomatic detection of preeclampsia in the first trimester remains to be demonstrated. Currently our pregnancy cohorts provide proof-of-principle in the prenatal setting and encourage large-scale efforts to annotate NIPS data with clinically actionable pregnancy outcomes. We

suspect that the approach could potentially be used to detect and explore the pathophysiology of a series of other adverse pregnancy outcomes, such as fetal growth restriction, congenital infections, and maternal autoimmune disease flares.

Importantly, our cfDNA-inferred cell type signatures are preserved at ultra-low sequencing depths. This is advantageous because such data is routinely generated in the prenatal and some oncology settings for clinical purposes, allowing for both disease mechanism exploration in clinical datasets and the rapid implementation of classifiers for patient stratification. Although highly specific, methylation and mutation-based methods require targeted capture and resequencing of informative fragments, resulting in longer and costlier processes in the clinic. A limitation of our approach is that the rank of a cell type may be spuriously elevated if it shares a transcriptomic profile with a contributing cell type. This is a limitation for most deconvolution approaches as there is typically some overlap between informative sites for related tissues and cell types. Despite potential collinearity between cell type features, our validation experiments show that we can identify rank changes that reflect real biological differences between males and females and pregnant and non-pregnant individuals. Our predictive analysis demonstrated the ability to differentiate high-coverage colorectal cancer (AUC 84.7%) and early-stage breast cancer (AUC 90.1%) from matched controls. Similarly, we distinguished low-coverage multiple myeloma (AUC 95.0%), and preeclamptic pregnancies (AUC 88.3%) from matched controls. Furthermore, our colorectal model successfully generalized to an external cohort of cfDNA samples sequenced at low-coverage (AUC 85.7%). Overall, the reported performance of existing classifiers varies significantly even within the same disease entity depending on the specific cfDNA metric used, the size and diversity of the study population, and the algorithm employed[12,51]. Notably, our classifier for early-stage breast cancer shows promise compared to existing fragmentation-based classifiers that we benchmarked against (i.e., ichorCNA, Griffin) and other published classifiers[30,52].

In summary, we utilize single-cell transcriptome atlases to rank a comprehensive set of potential cell type contributors to plasma cfDNA in health and disease. Changes in cfDNA contribution can indicate cell type loss, damage, or proliferation under (patho)physiological conditions. The approach's sensitivity at ultra-low sequencing depths enables cost-effective sequencing of large case-control cohorts for disease signature discovery and classification models. Using the approach, we see potential for the expanded clinical utility of standard-of-care prenatal screening (NIPS) and liquid biopsy data in cancer and diverse conditions.

## Methods

### Cell-free plasma DNA extraction, sequencing, and data preprocessing

The study was approved by the Ethical Committee of University Hospitals Leuven (study protocols S62285, S66450, S57999, S67127) and Vall d'Hebron Institute of Oncology (VHIO), Barcelona, Spain (study protocol PR(AG)321/2018). For the cfDNA samples prospectively collected, written informed consent was obtained. Patients that were retrospectively recruited were informed about their participation through a message accompanied by an information letter in their electronic health file application. Patients were excluded from the study in case they opted out. Peripheral blood samples from pregnant women, non-pregnant control samples, and cancer patients were collected in cell-free DNA collection tubes (Roche Diagnostics, Switzerland or Streck, USA). Plasma was separated by the standard dual centrifugation method. cfDNA was extracted using the QIAamp Circulating Nucleic Acid Kit (Qiagen Benelux B.V., Venlo, Netherlands) or automated Maxwell® HT ccfDNA Kit (Promega) according to the manufacturer's guidelines. Libraries for low-coverage sequencing were prepared using the KAPA HyperPrep kit (Roche Diagnostics) with IDT

adaptor ligation. Ultra-low-coverage (0.1–0.3-fold) whole-genome sequencing was carried out on the NovaSeq 6000 (Illumina, San Diego, CA, USA) generating 2 × 51 paired-end reads. Sequencing data was demultiplexed, quality checked, and adapters were trimmed using fastp (v0.12.4). Raw reads were aligned to the human reference genome hg38 using the Burrows-Wheeler aligner (v0.7.17). Duplicate reads were marked and removed using Picard (v2.18.23). Bams were sorted and indexed with samtools (v1.9). Libraries for breast and colorectal cancer cfDNA samples, as well as the remaining non-pregnant control samples were generated using the NEBNext Enzymatic Methyl-seq kit (New England Biolabs, Ipswich, MA, USA) following manufacturer's instructions. Libraries were quantified using Qubit dsDNA high sensitivity assay kit and Qubit 3.0 fluorometer (Thermo Fisher Scientific, Waltham, MA, USA) and sequenced at 10- to 35-fold coverage on the NovaSeq 6000 (Illumina, San Diego, CA, USA) using S4 flowcell generating 2 × 150 bp paired-end reads. Sequenced data was demultiplexed, quality checked and adapters were trimmed using fastp (v0.20), and then aligned to human genome hg38 using bwa-meth (v0.2.2).

## Window protection scores and nucleosome positioning analysis
We calculated window protection scores and performed nucleosome peak calling exactly as specified by Snyder et al. (2016) using source code from the shendurelab GitHub (https://github.com/shendurelab/cfDNA). The only difference was that we used BAM files aligned to reference genome hg38 instead of hg19. In brief, per-base window protection scores (WPS) were generated genome-wide using mono-nucleosomal fragments of cfDNA (120-180 base pairs). The WPS is a sliding window metric defined as the number of molecules spanning a 120 base pair window centered at a given genomic coordinate minus the number of molecules with an end-point within that same window. For nucleosome positioning analysis, a heuristic peak calling algorithm was applied to identify contiguous regions of high WPS indicative of nucleosome occupancy. UCSC liftOver was used to convert the Snyder et al. (2016) peak calls from hg19 to hg38 coordinates for comparison with peak calls from our data. We removed highly repetitive blacklisted regions from UCSC ("genomicSupDups") before calculating distances between peaks due to low mapping quality in these regions.

## Cell-of-origin analysis
For the cell-of-origin analysis we applied a fast Fourier transformation on smoothed WPS signals in the first 10 kilobase pairs (kb) of 19,536 genes with a consensus coding sequence (GRCh38) from Ensembl. The mean FFT intensity at the 193–199 frequency range was then calculated per gene. This was done using the shendurelab GitHub code without any parameter modifications. Snyder et al. (2016) correlated per-gene FFT-WPS with the bulk expression levels of those genes in 76 tissues and cell lines. Instead, we used single-cell RNA sequencing expression levels summarized per gene from the Tabula Sapiens database as a reference dataset. Cell types were grouped by 24 biopsied organs and 4 compartments (i.e., immune, epithelial, endothelial, stromal) based on annotations from the original publication. Cell types were ranked based on the strength of correlation with negative correlations receiving the highest rank.

## Preparation of the single-cell transcriptomic reference datasets
Single-cell RNA sequencing data was downloaded from Tabula Sapiens on CZ CELLxGENE Discover (https://cellxgene.cziscience.com/collections/e5f58829-1a66-40b5-a624-9046778e74f5). We downloaded "Tabula Sapiens – All cells" as an RDS file for import into R (v4.1.3) and then subset to data generated using the "10 × 3'" assay (n = 412,848 cells) to minimize batch effects. We also excluded germ cells (i.e., sperm). Data normalization had already been carried out and

stored in the object's data slot (object@assay$RNA@data) using the DataNormalization function. An additional metadata column "cell_type_tissue" was created by concatenating the "cell type" and "tissue_in_publication" metadata columns resulting in 456 unique cell type identifiers across the 24 biopsied organs. The AverageExpression function was used (group.by = "cell_type_tissue", layer = "data") to generate a data matrix with the average expression of genes grouped by the 456 cell type identifiers.

The Tabula Sapiens database does not include data from the placenta. For placenta, raw read count data and metadata files containing cell type annotations were downloaded from ArrayExpress for E-MTAB-6701. This included 8 unique cell types from the placental compartment of the maternal-fetal interface: syncytiotrophoblast, villous cytotrophoblasts, proliferative villous cytotrophoblasts, extravillous cytotrophoblasts, proliferative extravillous cytotrophoblasts, Hoffbauer cells, and two separate fetal fibroblast populations. The data matrix was read into R (v4.1.3) and a Seurat (v4.1.1) object was created using the CreateSeuratObject function. Cell-type annotations from the original publication were added as metadata to the Seurat object. Raw read counts were normalized using the NormalizeData function. The placenta data object was then integrated with the Tabula Sapiens data object using the IntegrateData function. The AverageExpression function was then used to calculate the average expression of the integrated data values per cell type across both datasets. The resulting gene x cell type data matrix was used for correlation with per-gene FFT-WPS in the healthy pregnancy cohorts. For the pregnancy complication cohorts, additional cell types from 15 other fetal tissues were added to the reference set using gene expression levels downloaded from the Fetal Cell Atlas (https://descartes.brotmanbaty.org/bbi/human-gene-expression-during-development/) using the same data normalization and integration procedure. This was done because we were interested in testing for non-placental fetal cell type signatures specifically in pregnancies with a vanishing twin (absorbed) – such an extensive reference dataset is not needed for most prenatal applications where non-placental fetal tissue is not expected to enter the maternal bloodstream.

**Benchmarking analysis.** For the bulk tissue transcriptomic reference we downloaded the consensus transcript expression levels summarized per gene in 50 tissues based on transcriptomics data from Human Protein Atlas (HPA) and GTEx (https://www.proteinatlas.org/download/rna_tissue_consensus.tsv.zip). We correlated per-gene FFT-WPS with gene expression levels and ranked the relative contribution of the 50 bulk tissues to cfDNA based on the strength of correlation. This is the same procedure we used for the cell-of-origin analysis using single-cell transcriptomic references. Tissue ranks were used as input features into a support vector machine with default hyperparameters using the same leave-one-out cross-validation procedure as we used for cancer prediction with cell type features.

For the ichorCNA analysis we cloned the git repository (https://github.com/broadinstitute/ichorCNA) and followed the ichorCNA wiki (https://github.com/broadinstitute/ichorCNA/wiki) for data processing. We used the readCounter function from HMMCopy to generate wig files for all cases and controls. IchorCNA tumor fractions were then estimated using the runIchorCNA.R script for cases and controls using the default panel of normal reference. For benchmarking against "Griffin" (Doelbey et al. 2022), we cloned the git repository (https://github.com/adoebley/Griffin_analyses) and followed the Griffin wiki (https://github.com/adoebley/Griffin/wiki) for data processing. We used the 30,000-sites files for 270 transcription factors as is suggested by the authors for generic cancer classification. Griffin calculates three metrics per transcription factor: central coverage, mean coverage, and

amplitude across all binding sites. Griffin therefore outputs 810 features per sample (3 × 270) which we used as a test dataset for their published LUCAS-trained model made available for future users (https://github.com/adoebley/Griffin_analyses/final_models/cancer_detection).

## Statistical analysis

For the case-control analysis, we used a two-sided Wilcox rank-sum test to compare the distribution of ranks for each cell type in cases versus controls using the rstatix package (v0.7.1). The Benjamini & Hochberg (BH) method was used to adjust $p$-values for multiple testing across 456 cell types (adult) in the cancer cohorts and 496 cell types (adult + fetal) in the pregnancy complication cohorts. Foldchanges were calculated for each cell type by taking the ratio of the changes between the mean rank of cases and the mean rank of controls over the mean rank of cases. We visualized the results using a Volcano plot using the ggplot2 (v3.4.2), data.table (v1.14.8), tidyverse (v1.3.2), ggrepel(0.9.3), and viridis (v0.6.2) packages in R (v4.1.3).

For the unsupervised analysis, principal component analysis (PCA) was used for dimension reduction, and PCs with eigenvalue > 1 (Kaiser's criterion) were extracted for distance matrix construction using the Euclidean distance, followed by Walktrap community detection to define clusters with fixed parameters (the nearest number of nodes was 3 with a walk step of 2) using the igraph package (v1.3.0). To visualize the dataset in lower dimensions, $t$-distributed stochastic neighbor embedding (tSNE) was used with the same PCs using the Rtsne package (v0.15). Clusters defined from the Walktrap community detection were used for tSNE annotation. We performed clustering and tSNE visualization for each disease cohort + matched controls using all 456 and 496 features as input for the cancer and preeclampsia cohorts, respectively.

For the classification task, cell type ranks were used as feature inputs to a support vector machine (SVM). We used ranks for 456 cell types (adult) and 496 cell types (adult + fetal) to classify cancer and pregnancy disorders from controls, respectively. We trained a linear kernel SVM with default hyperparameters (cost = 1) using the e1071 package (v1.7-12) in R (v4.1.3) and the pROC (v1.18) and viridis (v6.2) packages for visualization. No dimensionality reduction was performed for the cancer cohorts. Given the limited number of informative cell types for the pregnancy complication cohorts, principal component analysis (PCA) was performed using all cell type ranks and the top 10 PCs were used as input to the linear kernel SVM (cost = 1) for classification. Leave-one-out cross-validation was used to assess the performance of each model on the internal case-control cohorts.

For an external validation cohort of colorectal cancer patients, cfDNA was sequenced at ultra-low-coverage (<0.3-fold). To test the performance of our colorectal cancer model on external data, we built an SVM classifier using the entire internal dataset of colorectal cancer patients and controls (all sequenced at 10-fold coverage) using the e1071 package in R (v4.1.3). We used the predict function from the e1071 package to use this model to distinguish the external cases (<0.3-fold coverage) from internal <0.3-fold coverage controls. Using these predictions, we calculated the positive predictive value (PPV), the negative predictive value (NPV), the sensitivity, and the specificity of our colorectal model on the external test set.

For the multi-class classifier, we limited this analysis to breast and colorectal cancer as these samples were sequenced at the same depth along with matched controls. We used a one-vs-all approach by training two binary support vector machine classifiers with leave-one-out cross-validation and default hyperparameters. The first classifier treated breast cancer as the positive class and colorectal cancer + controls together as the negative class. The second classifier treated colorectal cancer as the positive class and breast cancer + controls together as the negative class. The classifier with the highest decision value was used to predict the final class label while all samples with a decision value below the cutoff for both classifiers were labeled as controls.

## Reporting summary

Further information on research design is available in the Nature Portfolio Reporting Summary linked to this article.

## Data availability

Whole-genome sequencing data generated in this study is under controlled access because patients did not consent to data deposition in public data repositories. Data is available through the European Genome-Phenome Archive under study accession number EGAS50000000178 upon request to the local UZ Leuven data access committee (dac@uzleuven.be) which checks informed consent form compliance and ensures that there are no legal impediments. Requests will be handled within a month. Conflicts are handled by an independent UZ Leuven data access committee advisory board. Publicly available single-cell RNA sequencing data were downloaded from Tabula Sapiens (https://tabula-sapiens-portal.ds.czbiohub.org/), Fetal Cell Atlas (https://descartes.brotmanbaty.org/bbi/human-gene-expression-during-development/), and first-trimester placenta (E-MTAB-6701). Bulk RNA sequencing data was downloaded from the Human Protein Atlas (HPA) and GTEx (https://www.proteinatlas.org/download/rna_tissue_consensus.tsv.zip). The hg38 reference genome was downloaded here http://hgdownload.soe.ucsc.edu/goldenPath/hg38/bigZips/hg38.fa.gz. Source data are provided with this paper.

## Code availability

Code for data processing is available at https://github.com/shendurelab/cfDNA. Custom code for the case-control and predictive analyses can be accessed at https://github.com/JorisVermeeschLab/cfDNA_cell_of_origin.

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

## Acknowledgements

We would like to thank the patients and their families for agreeing to participate in this study. We would like to thank the Genomics Core at KU Leuven for the use of the sequencing facilities. Funding was received from the European Union's Horizon 2020 research and innovation program under grant agreement No 824110 – EASI-Genomics (J.R.V.) and the Marie Skłodowska-Curie grant agreement No 813707 (MATER), from Kom Op Tegen Kanker (Stand up to Cancer) KOTK/2018/11468, from the Flemish cancer society (2016/10728/2603 to A.C.), and FWO-SBO grant S003422N, and from Agentschap Innoveren en Ondernemen (VLAIO; Flanders Innovation & Entrepreneurship grant HBC.2018.2108). S.T. was supported by FWO SB/1S74420N. Institutional support was received from the KU Leuven, C1-C14/18/092, C14/22/125, and C3/20/100 to J.R.V.

## Author contributions

K.E.S and J.R.V. designed and planned the study. K.E.S. and J.R.V. wrote the manuscript. K.E.S. performed all the data analysis and developed the bioinformatic approach. K.E.S., T.J., B.T. and J.R.V. interpreted the data. S.T. performed sequencing experiments. S.T., D.S., L.L., K.V.C., M.B., I.V.P., L.V.C., K.V.B., R.A.T., L.Le., S.Te., K.P., L.Y.R. and P.V., organized patient enrollment, sample collection, and clinical data curation. All authors read and approved the final version of the manuscript.

## Competing interests

The authors declare no competing interests.
