## [Peer Review File · Nature Communications]

REVIEWER COMMENTS

Reviewer #1 (Remarks to the Author): expert in prenatal cfDNA analysis

This paper describes the use of single cell transcriptome data to analyse cell / tissue of origin of cell-free DNA. This is then correlated with disease characteristics and potential application in oncology and prenatal care. I feel that this is a very important seminal paper, and the authors should be congratulated on the level of work and amount of data in the paper. I suspect it will hold importance in the increasing use of liquid biopsy in several areas of medicine, and could possibly justify several publications with specific focus on clinical area.

I felt that the paper is well-written and requires very little editing.

It would be helpful to have more clinical information on the patient demographics and timing of the sampling with respect to the pregnancy / tumour development etc.

My clinical focus is prenatal - it would have been helpful to have more information such as singleton / twin pregnancy, patient demographics, gestation at sampling with respect to development of pre-eclampsia / miscarriage etc. Lines 283-287 - ie what gestation was there evidence of a twin, was there a fetal pole, and was NIPS data after that. With miscarriages did they all happen later in pregnancy after NIPS or were some due to missed miscarriage. Did you look at any early losses with placenta still in situ. With the pre-eclampsia cohort, was there any NIPS data from first trimester, or only at time of clinical diagnosis. There is variability of fetal fraction with maternal obesity, smoking, ethnicity, chorionicity, drugs etc and it would be interesting to explore this.

I agree that there is great potential in this methodology to provide novel insights into disease biology and to facilitate some of our current methods.

It is also possible that it could be developed for more advanced clinical use in future but there needs to be much larger cohorts with more detailed clinical information in specific conditions with paired control groups (paired for demographics too) and demonstrating clinical utility.

Line 407-408: the authors state that they "envision the expanded clinical utility of SOC NIPS and liquid biopsy data based on this technology", but I think in my opinion this is a bit of a leap based on the data in this paper alone. How do the authors think that this analysis may be integrated into standard NIPS screening? Is there an intention to look in more detail for example with pre-eclampsia, to see how early the signatures are evident? It's not so helpful being able to see the signatures at a time when a patient is already clinically diagnosed.

Are there other pathologies where this may be relevant eg looking at fetuses with IUGR, hydrops, neurological conditions, neural tube defects. Do you expect that it may improve the performance of NIPS detection of T21 / 13/ 18 by detecting specific signatures?

It is exciting that this may be a way forward with developing testing, but I suspect this is maybe a way off yet?

There is a huge amount of information included in the figures and it is well-represented and explains the concepts well. Care must be taken in the final version to make sure that everything in the published figures is visible as parts are very small, and that the colours used are distinguishable.

Fig2A - axis labelling is difficult to see

Fig 5 A and C - I couldn't easily distinguish the different coloured points in the graphs, and I found it confusing as to which points the writing on the graphs was referring to.

Line 93 - needs a numbered link to reference

In conclusion - a well-written and exciting paper worthy of publication with very minor modification!

Reviewer #2 (Remarks to the Author): expertise in low pass whole genome sequencing and cfDNA

In the current manuscript, Stanley and colleagues develop and validate a method for disease-agnostic cell free DNA cell-of-origin analysis. The method leverages cfDNA fragmentation patterns within gene bodies to identify a signature that is correlated with expression in the cell of origin. By correlating these signatures with single cell expression profiles from the Tabula Sapiens and other datasets, the authors produce a list of cell types ranked by similarity to each sample. The authors

demonstrate that the relative position of cell types in the ranked list is biologically relevant, and can be applied in tasks such as classification and biomarker discovery.

The authors validate their method by demonstrating robustness across input sequencing depths down to 0.3x coverage. They validate that certain sex-specific cell populations are up/down ranked in male/female controls, and observe high ranking of pregnancy-associated cell populations in pregnant individuals. The authors then apply their method to case-control analyses in three different cancers: colorectal, breast, and multiple myeloma. The observed expected up-ranking of disease associated cell populations in colorectal and multiple myeloma, which positively correlated with ichorCNA estimated tumor fraction, but did not observe such associations in their breast cancer cohort, which they attributed to a lack of later-stage (and therefore high tumor content) samples.

The authors demonstrated the predictive performance of their method by using the ranks of cell populations to train a linear SVM for disease vs control classification and apply their trained colorectal cancer model to an external test set. Finally, the authors demonstrate the ability of their tool to identify biologically relevant cell populations that are up-ranked in preeclampsia, providing insight into disease biology.

Overall the paper is well written and the results are clearly presented. The statistical analyses are reasonable. While the approach here is similar to previous studies utilizing fragmentomics, this does seem like a step forward, building on prior studies. I do have the following questions for the authors.

1. Given the existing literature around cancer subtype classification using fragmentomics (both targeted and WGS), what is the authors' view as to the novelty of going through cell-type intermediaries to get to the same eventual conclusion? That is to say, if the end-goal is cancer classification (or cancer / no-cancer classification), what benefit is accomplished by correlating with cell-type ranks?
2. Regarding the correlation with cell-type specific features, it seems that many of these could be highly correlated and/or overlapping. For example, in Table S1 for the "colorectal cancer" classifier, the top "epithelial" cell types that are enriched are "basal_cell_trachea", "club_cell_lug", "epithelial_cell_prostate". It seems problematic to have these be ranked higher than anything of GI origin for colon cancer when considering a potential multi-tissue-of-origin classifier.
3. Regarding the most enriched cell-types in healthy control cfDNA (Fig 2F), there is a significant and interesting discrepancy in the cell top cell-types identified here compared with a recent study using methylation to deconvolve healthy cfDNA (Loyfer et al, Nature 2023). In this study using methylation profiles from sorted cells, there is very little contribution in the cfDNA space from lymphocytes (2% of all molecular contribution). However here, after monocytes, the next 6 most enriched cell types are all lymphocytes (NK cells, T cells, and thymocytes), with many more lymphocyte populations enriched. What do you think leads to this discrepancy?
4. Overall, it does seem like this system works, although largely works at higher tumor fractions. What do the authors think is the limit of detection for robust classification of, 1) cancer versus no-cancer or, 2) specific cell types of interest?

Minor comments

5. Minor errors:

o Figure 4.G describes AUC confidence intervals not on the figure.

o Please specify FDR correction method (BH?)

o Figures seem mislabeled – for example there is reference to Figure 4J but I do not see Figure 4J.

6. What do the authors think is the best use-case for this method? Patient identification? (ie, cases versus controls) Biological discovery? Or is this a methods paper?

7. Benchmarking classification versus alternative methods that can be performed on WGS would be helpful. Is classification / deconvolution truly improved?

8. Regarding the cancer classification portion of the text – can the authors comment on the most important features for classification? Do these make biological sense?

Reviewer #3 (Remarks to the Author): expertise in cfDNA fragmentation bioinformatics

Stanley et al. attempted to demonstrate that cell type signatures in cell-free DNA fragmentation profiles could reveal disease biology. They utilized single cell transcriptome data as a comprehensive cellular reference set to facilitate cfDNA cell-of-origin analysis. They correlated cfDNA-inferred nucleosome spacing with gene expression to rank relative contribution of over 490 cell types to plasma cfDNA. Based on this information, a predictive model was trained for differentiating patients with colorectal cancer, early-stage breast cancer, multiple myeloma and preeclampsia, with an AUC of at least 85%. This is an interesting study. Below are the comments which the authors should address.

1. What is the main advantage of using signatures identified from the single cell RNA sequencing, comparing to the previous study by Doebley et al. published the same journal, titled "A framework for clinical cancer subtyping from nucleosome profiling of cell-free DNA"?
2. For classification task based on support vector machine, the input features comprising 456 cell types to classify cancer. However, the sample size ($n = 16$ and 52 , for colorectal cancer and breast cancer) seems very limited relative to the number of features. Hence, the actual performance might be still unanswered. It will be informative to use unsupervised methods such as PCA or tSNE to visualize the data structure and check whether the patients will be clustered together depending on the pathophysiologically states using these features.
3. It will be interesting to explore what is the minimal number of cell types required for classification of cancer with a desired performance. From this study, it is not clear to the reviewer as to how much improvement in window protection score based cfDNA nucleosome spacing analysis is contributed by the use of single cell RNA sequencing, if compared with the traditional bulk RNA sequencing.
4. In Fig. 2D, both uterus and prostate showed significant ranks, for example, above intestines. Did author compare these tissue contributions (uterus and prostate) between female and male subgraphs?

Reviewer #4 (Remarks to the Author): expertise in colorectal cancer genomics with single cell

Review comment

In the manuscript "Cell type signatures in cell free DNA fragmentation profiles reveal disease biology", the authors have presented a method that employs single cell transcriptome data to analyze the origin of circulating cell-free DNA (cfDNA). This approach demonstrates the ability to effectively differentiate patients with various diseases, including colorectal cancer, early-stage breast cancer, multiple myeloma, and preeclampsia, even in low-coverage cfDNA datasets. The significance of the study relies on two key aspects; its novelty and the performance of the method compared with previous methodologies. I find the use of single cell transcriptome data for cfDNA fragmentome analysis is both timely and interesting. However, it's worth noting that the study lacks comparisons to other cfDNA analyses (e.g. methylation), which are essential to anticipate the potential applications of this research.

Major comment

1. The authors acknowledged that this study builds on the work by Snyder et al (2016) by utilizing single cell transcriptome atlases instead of bulk tissue references. The method descriptions for WPS and FFT with respect to gene expression are relatively brief. Please elaborate on any differences from Snyder et al.'s methods, if present, clearly indicating what has been modified, why these modifications were made, and the resulting benefits.
2. Preparation of the Single cell transcriptome atlas should be described more in detail for the readers to repeat the process.
3. Prediction sensitivity and specificity are known to differ depending on the study population. It would be good to use other cfDNA analysis (methylation etc) to compare performance.
4. Specific comments for the figures
Fig 2G shows the down sampling results for the highest and lowest. What about other cell types? What about cell types used for the classification experiments?
Fig 3C : which specific cell types within placenta?

Fig 3D : what about the rank of other cell types?
Fig 3F: Statistics?
Fig 4F: Stage of multiple myeloma?
Fig 4B,D,F: what are the ranges of healthy controls?

RESPONSE TO REVIEWERS' COMMENTS

We thank the reviewers for their careful reading of our manuscript and for providing suggestions that we think have greatly improved the manuscript. Below, we have included a summary of the key revisions, followed by point-by-point responses to reviewer comments.

Summary of key revisions

1. Further benchmarking against other methods, suggested by Review 2 + 4.

We have performed benchmarking against bulk tissue transcriptomic references (Snyder *et al*, 2016), and two additional fragmentation-based cfDNA methods for cancer detection called ichorCNA (Meyerson *et al*, 2017) and Griffin (Doebley *et al*, 2022).

We decided to benchmark against bulk tissue transcriptomic references because both Reviewer 1 + 2 questioned the added value of using cell type features as opposed to tissue features for cancer prediction. In our cohorts, using bulk tissue features decreased the AUROC from 95.0% to 87.1%, 84.7% to 73.4%, 90.2% to 84.6%, and 88.3% to 61.5% for multiple myeloma, colorectal cancer, breast cancer, and preeclampsia, respectively. IchorCNA uses binned genome-wide cfDNA fragment coverage to estimate tumor fraction in cancers with copy number alterations. In our cohorts, using ichorCNA decreased the AUROC from 95.0% to 92.6, 90.2 to 65.7, and 84.7% to 64.2% for multiple myeloma, breast, and colorectal cancer, respectively, highlighting the major advantage of our method in patients with low tumor burden or copy number neutral cancer. We then benchmarked against Griffin as it does not rely on copy number alterations and has been demonstrated to perform well in early-stage cancers sequenced at low coverage. Griffin uses normalized fragment coverage at the binding sites of 270 transcription factors to capture changes in gene regulation for cancer prediction. In our cohorts, Griffin achieved an AUROC of 75.6%, 77.4%, and 90.9%, in multiple myeloma, breast, and colorectal cancer, respectively. Please refer to point-by-point responses for details.

2. PCA and/or tSNE analysis as suggested by Reviewer #3

We have performed an additional set of exploratory analyses using unsupervised methods. Principal component analysis (PCA) was performed using all 456 or 498 cell type features for cancer or preeclampsia, respectively, followed by Walktrap community detection to define clusters and t-distributed stochastic neighbor embedding (tSNE) for visualization. We are encouraged to see that even in the unsupervised space there is uneven distribution of cases and controls in the walktrap community clusters and apparent separation between cases and controls in the tSNE visualization. We think these analyses better define the data structure and support the validity of our supervised models. Please refer to the point-by-point responses for details.

Point-by-point responses:

Reviewer #1 (Remarks to the Author): expert in prenatal cfDNA analysis

1. It would have been helpful to have more information such as singleton / twin pregnancy, patient demographics, gestation at sampling with respect to development of pre-eclampsia / miscarriage.

We have added “TableS6_MetaData_Pregnancy” with the information that Reviewer #2 requested if it was available to us. This includes the chorionicity, the gestational age at sampling, the gestational age at preeclampsia diagnosis for the preeclampsia cases (same as gestational age at sampling), and the gestational age of fetal demise for the miscarriage cases.

2. Lines 283-287 - ie what gestation was there evidence of a twin, was there a fetal pole, and was NIPS data after that.

For the vanishing twin cohort, the presence (yes/no) of a vanishing twin is encoded in our retrospective NIPS database as it can aid in the interpretation of NIPS results for chromosomes 13, 18, and 21 in the standard clinical workflow. We do not have information on what ultrasound findings were used to confirm the presence of a vanishing twin (e.g. dark spot, yolk sac or embryo visible). As mentioned in the discussion lines 368-370, we think the variability we see in this cohort may be due to the exact timing of the twin’s demise or due to falsely annotated vanishing twins. Detailed analysis of ultrasound images by a specialist and selection of cases that have higher levels of evidence (e.g. yolk sac / embryo visible) could potentially improve the classification of vanishing twin using our approach. For all cases, blood sampling for NIPT was performed after the demise of the vanishing twin, so we’re detecting the post-hoc event rather than predicting fetal demise.

3. With miscarriages did they all happen later in pregnancy after NIPS or were some due to missed miscarriage.

At our center, NIPT is not typically performed if a miscarriage occurs before the time of NIPT blood sampling (around 10-14 weeks) hence by definition all happened after or at the time of sampling. The exact gestational age of fetal demise was available in our retrospective NIPT database for 29 of the 44 miscarriages. This information is now included in Table S6 along with the gestational age at sampling. For most cases there is several weeks between fetal demise and NIPT sampling. For two cases the miscarriage was recorded at the time of NIPT sampling.

4. Did you look at any early losses with placenta still in situ.

Unfortunately, we do not have this level of ultrasound information for our pregnancy loss cohort. Many of the NIPT samples at KU Leuven are referred to us by external hospitals and so that granularity of information is not usually available to us because the pregnancies are not seen at KU Leuven or followed up here.

5. With the pre-eclampsia cohort, was there any NIPS data from first trimester, or only at time of clinical diagnosis.

Only at time of clinical diagnosis.

6. There is variability of fetal fraction with maternal obesity, smoking, ethnicity, chorionicity, drugs etc and it would be interesting to explore this.

Our retrospective NIPT database does not include information on patient ethnicity, obesity, smoking status, or drug use. We agree that it would be very interesting to investigate to what extent these variables impact fetal fraction and more generally cell type contributions to cell free DNA. We were able to go back and gather this information for the 18 preeclampsia cases and 30 gestationally age-matched controls that were collected prospectively and have included this information in Table S8. We have now checked that the preeclampsia cases and controls are not only matched for gestational age but also for maternal BMI, maternal age at conception, smoking status, and ethnicity (Table S8, Figure S11, added text lines 324-326). We thank Reviewer #2 for their comment as we think this additional information addresses potential confounding factors and supports that our findings are preeclampsia-specific nature of our findings.

7. Line 407-408: the authors state that they "envision the expanded clinical utility of SOC NIPS and liquid biopsy data based on this technology", but I think in my opinion this is a bit of a leap based on the data in this paper alone. How do the authors think that this analysis may be integrated into standard NIPS screening?

We agree that the language is a bit strong and have modified the text to read “Using the approach, we see potential for the expanded clinical utility of standard-of-care prenatal screening” (line 462). One of the major potential advantages of this approach is that, unlike cfDNA methylation-based approaches, this approach would not require any changes to current clinical NIPS pipelines to be trialed and implemented in Belgium and the Netherlands, where whole-genome paired-end NIPS is already routinely performed in over 80% of the population. Any predictive model based on our cell type signatures could be run with very minimal additional computational costs and added as a flag to the standard NIPT report. We emphasize this advantage in the discussion lines 435 - 437. At centers that perform targeted single-end sequencing of chromosomes 13, 18, and 21 for NIPT, a switch to genome-wide paired-end sequencing would be required.

8. Is there an intention to look in more detail for example with pre-eclampsia, to see how early the signatures are evident? It's not so helpful being able to see the signatures at a time when a patient is already clinically diagnosed.

Yes, the hope is that the approach will be applied on clinically actionable pregnancy complication cohorts sampled early on in gestation. This includes presymptomatic preeclampsia cases sampled early on in gestation which have been shown recently to have altered cfDNA methylation profiles¹. We think this is a particularly important potential case use as aspirin must be administered before 16 weeks' gestation to have a therapeutic effect. We have added lines 426 - 428 to the discussion to emphasize this point. Our preeclampsia cases were sampled in the second trimester at diagnosis (24-34 weeks' gestation; this is described in the text lines 322 – 323 and also now detailed in Table S7-8). Large-scale prospective studies would also need to be performed to assess the true performance of any predictive model using our cell type signatures in the prenatal setting.

9. Are there other pathologies where this may be relevant eg looking at fetuses with IUGR, hydrops, neurological conditions, neural tube defects.

We envision that this approach could be used for risk stratification in, for instance, pregnancies with maternal autoimmune disorders or maternal infection which may or may not result in pregnancy complications. Also, as Reviewer #1 suggests, we suspect that the approach could potentially be used for the early screening of other adverse pregnancy outcomes such as IUGR, hydrops, or neural tube defects, if there is a maternal immune response or placental involvement that can be detected in the pool of maternal cfDNA. We have added some of these potential use cases to the discussion lines 430-432.

10. Do you expect that it may improve the performance of NIPS detection of T21 / 13/ 18 by detecting specific signatures?

We do not necessarily expect this approach to improve the detection of T21, T13, T18 directly. It is well established that read counting statistics used during routine NIPT can detect increased mapping of fetal reads to specific chromosomes affected by trisomy, but whether this results in a robust change in cell types contributing to cell free DNA is less clear. Should there be increased or decreased turnover of placental or maternal immune derived cells due to the aneuploidy-associated phenotypes (e.g. growth restriction), then our method may be able to capture this. This is certainly an interesting area for future exploration.

11. It is exciting that this may be a way forward with developing testing, but I suspect this is maybe a way off yet?

We agree that the clinical utility of this approach in the prenatal setting remains to be demonstrated in larger cohorts with more detailed clinical annotations. Currently our prenatal cohorts provide proof-of-principle in the prenatal setting and encourage large-scale efforts to annotate NIPT data with clinically actionable pregnancy outcomes. We have added a sentence to the discussion emphasizing this point (lines 427-429). We agree that this is a very exciting way forward and justifies several publications with a specific clinical focus. It is however beyond the scope of the current study which primarily aims to validate the approach and to demonstrate the increased resolution of biological signals that can be captured in cfDNA fragmentation profiles for a wide range of disease biology exploration.

Fig2A - axis labelling is difficult to see

We have increased the size of the x-and y-axis labels.

Fig 5 A and C - I couldn't easily distinguish the different coloured points in the graphs, and I found it confusing as to which points the writing on the graphs was referring to.

We have added lines connecting the cell type labels and colored points.

Line 93 - needs a numbered link to reference.

We thank Reviewer #1 for catching that, we have added the numbered reference.

Reviewer #2 (Remarks to the Author): expertise in low pass whole genome sequencing and cfDNA

1. Given the existing literature around cancer subtype classification using fragmentomics (both targeted and WGS), what is the authors' view as to the novelty of going through cell-type intermediaries to get to the same eventual conclusion? That is to say, if the end-goal is cancer classification (or cancer / no-cancer classification), what benefit is accomplished by correlating with cell-type ranks?

We thank Review #2 for their comment. We think one of the major benefits of capturing specific cell type signatures in cfDNA data is that our features are highly interpretable for clinical

specialists and can be used to explore specific cell types of interest, more so than other less biologically interpretable cfDNA metrics (e.g. fragment size, genome-wide fragment coverage). For example, the infiltration of specific immune cell types into the tumor microenvironment could be traced across treatment regimens to monitor response. These notions are in line with recent work by the Vogelstein lab². Our intestinal CD8⁺ T cell signal in colorectal cancer demonstrates that it is possible to track these kinds of cell type dynamics. So far, it is our opinion that other approaches (e.g. ichorCNA, Griffin) that use bulk tissue references are less well suited to answer these types of biologically oriented questions in oncology and prenatal care. We have added this advantage to the discussion lines 382-385. We have also demonstrated in an additional benchmarking exercise that using higher resolution cell type features, rather than bulk tissue features, boosts cancer and preeclampsia prediction performance in our cohorts (please refer to “Summary of key revisions” and to point 7 below for details on the benchmarking exercise).

2. Regarding the correlation with cell-type specific features, it seems that many of these could be highly correlated and/or overlapping. For example, in Table S1 for the “colorectal cancer” classifier, the top “epithelial” cell types that are enriched are “basal_cell_trachea”, “club_cell_lug”, “epithelial_cell_prostate”. It seems problematic to have these be ranked higher than anything of GI origin for colon cancer when considering a potential multi-tissue-of-origin classifier.

We agree that correlation between cell type features should be considered during data interpretation. The rank of a cell type may be spuriously elevated if it has a transcriptomic profile similar to another contributing cell type. We would like to note that this is a limitation for most deconvolution approaches as there is typically some overlap between informative sites for related tissues and cell types. Our validation experiments show that we can identify rank changes that reflect real biological differences between males and females and pregnant and non-pregnant individuals despite potential collinearity between cell type features. For instance, uterine cells still receive a correlation value and rank in males, but are significantly higher in females where the organ is actually present. Vice versa for prostate cells. Placental cells also receive a baseline correlation value in non-pregnant individuals where the organ is not present, but those values are significantly stronger in pregnant individuals where the organ is present. We have added this as a limitation to the discussion (lines 439-445).

As Review #2 notes, if the cell type signatures were to be used in a multi-class classification task and if there was an overlap between cell type associations then model performance might be impacted. To address Reviewer #2’s concern, we tested the performance of the cell type features in a multi-class setting using a one-vs-all support vector machine with default hyperparameters. We limited the analysis to breast and colorectal cancer patients + controls as they were sequenced at similar depths. We trained two binary classifiers with leave-one-out cross validation. The first classifier treated breast cancer as the positive class and colorectal cancer +

controls together as the negative class. The second classifier treated colorectal cancer as the positive class and breast cancer + controls together as the negative class. The classifier with the highest decision value was used to predict the final class label (breast or colorectal) while all samples with a decision value below the cutoff for both classifiers were labelled as controls. The one-vs-all SVM was able to correctly classify the majority of both breast cancer patients (44/52) and colorectal cancer patients (6/16) with an overall accuracy of 71.0. Of the misclassified colorectal cancer patients, 5/16 (31%) were predicted as breast cancer and 5/16 (31%) were predicted as controls. We think this begins to demonstrate that there are enough biologically meaningful and distinct cell type associations to classify our cancer patients even when considering them all together. We thank the reviewer for this interesting suggestion and have added decision values, predicted class, and contingency tables are in “TableS5_One-vs-all-SVM” and a description of these results are now included in the results section lines 310 -316 and the methods section 619 - 626.

3. Regarding the most enriched cell-types in healthy control cfDNA (Fig 2F), there is a significant and interesting discrepancy in the cell top cell-types identified here compared with a recent study using methylation to deconvolve healthy cfDNA (Loyfer et al, Nature 2023). In this study using methylation profiles from sorted cells, there is very little contribution in the cfDNA space from lymphocytes (2% of all molecular contribution). However here, after monocytes, the next 6 most enriched cell types are all lymphocytes (NK cells, T cells, and thymocytes), with many more lymphocyte populations enriched. What do you think leads to this discrepancy?

It is difficult to make one-to-one comparisons with Loyfer *et al.* because we rank the relative contribution of cell types rather than estimate their absolute proportions. The ranking should be interpreted as the next most abundant cell type. That said, we agree with Reviewer #2 that based on their analysis one would expect granulocytes to rank more higher than lymphocytes and monocytes. We speculate in the discussion (line 396) that NETosis, a neutrophil-specific form of cell free DNA release, may account for this discrepancy by reducing the proportion of mono-nucleosomal (120-180bp) fragments needed for our analysis. It is known that NETs play an important role in pathologies such as systemic lupus erythematosus where cfDNA fragment size profiles are known to be significantly altered².

In attempt to resolve this issue we performed additional analysis without fragment size selection to test for the presence of neutrophil-derived DNA in our data. To do this we calculate the aggregated mean fragment coverage in the ± 5 kb flanking region of transcriptional start sites (TSS) predicted by chromHMM for neutrophils, monocytes, T cells, liver, and placenta in healthy first trimester NIPT data (see additional figure below). In this analysis we indeed see the neutrophil signature is strongest, as suggested by Loyfer *et al.*, followed by monocytes, placenta, T cells and liver cells.

This analysis rules out technical errors related to sample preparation that may have accounted for the discrepancy. Based on the current literature it is not yet clear to what extent fragment size distributions are altered in NETosis versus apoptosis, and more work is needed to confirm this speculation. It is clear our current analysis has a reduced ability to measure neutrophil-derived cfDNA despite them contributing. We have highlighted this discrepancy in the text but have not included the additional analyses in the manuscript, as we feel it overly complicates the discussion and further research is needed to fully address the limitation.

It is also worth noting that T-lymphocytes exhibit a greater subtype diversity compared to other immune cell types *in vivo*. For this reason, we have a larger number of T-lymphocyte populations included in our reference set compared to say monocytes, granulocytes, and B-lymphocyte populations. It is therefore problematic to compare the number of T-lymphocyte subtypes to the number of other immune cell subtypes in the ranking as Reviewer #2 does, “here, after monocytes, the next 6 most enriched are lymphocytes”. We find it interesting that B lymphocytes appear among the top ranked cell types along with T lymphocytes. This corroborates recent reports^{3,4} of a large discrepancy between the 4:1 ratio of B to T cells expected in circulation and the 1:1 ratio of B cell-derived DNA to T cell-derived DNA in circulation. We’ve added this insight and references to the discussion lines 398 – 401.

4. Overall, it does seem like this system works, although largely works at higher tumor fractions. What do the authors think is the limit of detection for robust classification of, 1) cancer versus no-cancer or, 2) specific cell types of interest?

We demonstrate that the system works to distinguish cancer versus no-cancer at both higher and lower tumor fractions – as is the case for the early-stage breast cancer cohort. We think this is one of the major potential benefits of the approach. Many cfDNA approaches for cancer detection quantify tumor-derived copy number changes (e.g. ichorCNA) or use pre-selected tumor-specific markers. For this reason, they do not capture immune-related signals during early cancer dissemination when signals from the primary affected tissue / tumor are more difficult to detect. Currently, we can say that the limit of detection is approximately 0.01 which is the median tumor fraction of our early-stage breast cancer cohort, but again the analysis is not solely designed to identify tumor cfDNA so the limit of detection of cancer as it is usually understood in liquid biopsy may not be as relevant.

We thank Reviewer #2 for their suggestion to comment on the limit of detection for other specific cell types. We think this is a pertinent question. In our validation experiments there is a significant difference in the rank of sex-specific cell types between males and females. We assume these cell types (i.e. uterus, prostate) contribute minimally to cfDNA populations, although their absolute proportions are not estimated here and have never been estimated using alternative methods. Given the sex-specific contributions we note in the validation experiments among healthy individuals, we wanted to make sure that our cancer models were predicting cancer and not sex. We have now matched the cancer cohorts to controls by sex. For colorectal cancer and multiple myeloma, the cases and controls are already matched for sex so the results have not changed (see new Table S1). For the breast cancer cohort, we have now limited the controls to females (n=88). We are encouraged to see that this did not greatly impact the results. We identified similarly up- and down-ranked cell types in the case-control analyses, notably macrophages being the most differentially ranked cell types between cases and controls (Table S3 now reflects these changes). The performance of our predictive model for breast cancer decreased marginally from an AUC of 92.3 to 90.1 (Figure 4G now reflects these changes). We have also specified that cases and controls are matched for sex in the text lines 220 and 238 and point to Table S1 where we test for differences in sex demographics between cases and controls for each cancer cohort using a Fisher's exact test.

Minor comments

5. Minor errors:

o Figure 4.G describes AUC confidence intervals not on the figure.

We thank Reviewer #2 for catching that, we have added the confidence intervals.

o Please specify FDR correction method (BH?)

We have changed the text and figure to Benjamini & Hochberg (1995) (“BH”) which was the method used.

o **Figures seem mislabeled – for example there is reference to Figure 4J but I do not see Figure 4J.**

We thank Reviewer #2 for catching that, we have corrected the text to Figure 4H.

6. What do the authors think is the best use-case for this method? Patient identification? (ie, cases versus controls) Biological discovery? Or is this a methods paper?

We think this is primarily a methods paper, although our applications in the cancer and prenatal cohorts do provide a new line of evidence for emerging disease paradigms which supports biological discovery. We think there are very exciting use cases in the prenatal field where these types of cell-of-origin analyses have not yet been thoroughly applied. Disease biology for many obstetric conditions is understudied and remains unknown. In the prenatal setting, our method may primarily be used for biological discovery. Please refer to our responses to Reviewer #1 (point 8 + 9) for details on potential applications in the prenatal setting (e.g. maternal autoimmune disorders / maternal infection / fetal growth restriction). We have also added these potential use cases to the discussion lines 430-432. Given the relative plethora of biomarkers in oncology, it may be that the best use cases in oncology focus on the rank of specific cell types of interest, now mentioned in the discussion 382 - 385. We have mentioned some use cases above in oncology where we think this method is well suited (e.g. tracking immune cell infiltration across treatment regimens to monitor response).

7. Benchmarking classification versus alternative methods that can be performed on WGS would be helpful. Is classification / deconvolution truly improved?

We thank Reviewer #2 for the suggestions and agree that benchmarking is an important exercise. We have performed benchmarking against bulk tissue transcriptomic references (Snyder *et al*, 2016), ichorCNA (Meyerson *et al*, 2017), and Griffin (Doebley *et al*, 2022).

For the bulk tissue transcriptomic references, we downloaded the consensus transcript expression levels summarized per gene in 50 tissues based on transcriptomics data from Human Protein Atlas (HPA) and GTEx (https://www.proteinatlas.org/download/rna_tissue_consensus.tsv.zip).

These tissues largely correspond to the tissues from the Tabula Sapiens database for which we have single cell data and include placenta. This is the updated version of the file that Snyder *et al* used in their analysis in 2016 (<http://www.proteinatlas.org/download/rna.csv.zip> -- no longer available). In our cohorts, using bulk tissue features instead of cell type features decreased the AUROC from 95.0% to 87.1%, 84.7% to 73.4%, 90.2% to 84.6%, and 88.3% to 61.5% for multiple myeloma, colorectal cancer, breast cancer, and preeclampsia, respectively. We have added ROC curves for the bulk tissue models to Figure S8 and an explanation to the main results section lines 287 – 294 and the methods section lines 556 - 564.

For the ichorCNA analysis we cloned the git repository (<https://github.com/broadinstitute/ichorCNA>) and followed the ichorCNA wiki (<https://github.com/broadinstitute/ichorCNA/wiki>) for data processing. We used the readCounter function from HMMCopy to generate wig files for all cases and controls. IchorCNA tumor fractions were then estimated using the runIchorCNA.R script for cases and controls using the default panel of normal reference. IchorCNA was able to distinguish between cases and controls with poorer performances compared to our cell type features with AUCs of 92.6 for multiple myeloma, 64.2% for colorectal cancer, and 65.7 for breast cancer. This highlights the major advantage of our method in patients with low tumor burden or copy number neutral cancers. We have added ROC curves for ichorCNA to Figure S9 and an explanation to the main results section lines 296 – 301 and the methods section lines 566 - 571.

We have also performed benchmarking against fragmentation-based cfDNA method “Griffin” (Doelbey et al, 2022) as it does not rely on copy number alterations and has been demonstrated to perform well in early-stage cancers sequenced at low coverage. Griffin instead uses 3 fragment coverage metrics at the binding sites of 270 transcription factors to capture gene regulatory information. Also for this analysis, we cloned the git repository (https://github.com/adoebley/Griffin_analyses) and followed the Griffin wiki (<https://github.com/adoebley/Griffin/wiki>) for data processing. We used the 30000-sites files for 270 transcription factors as is suggested by the authors for generic cancer classification. Griffin output 810 features per sample (3x270) which we used as a test dataset for their model. Griffin performed better than our cell type features in the context of high-coverage later-stage colorectal cancer (AUROC 90.9% compared to 84.7%) but worse for high-coverage early-stage breast cancer (AUROC 77.4% compared to 90.2%) and for low-coverage later-stage multiple myeloma (AUROC 75.6% compared to 95.0%). Overall, we think Griffin is a useful framework, but has some limitations that our cell type features do not. A major difference is that Doelbey *et al* do not present a way to utilize their features for tissue-of-origin analysis and cannot be used to determine the location of tumors or involved cell types. We have added ROC curves for the Griffin model to Figure S10 and an explanation to the main results section lines 301 – 308 and the methods section lines 571 - 578.

8. Regarding the cancer classification portion of the text – can the authors comment on the most important features for classification? Do these make biological sense?

We extracted the feature coefficients from the SVM models to check which cell types were most important for classification. Unsurprisingly, we observed overlap between important SVM features and those highlighted in the case-control analysis for each cancer cohort (Supplementary Tables S2-4 and S9-11). In particular, B cells and plasma cells in multiple myeloma, intestinal immune cells in colorectal cancer, and macrophages and other immune cell types in breast cancer.

Reviewer #3 (Remarks to the Author): expertise in cfDNA fragmentation bioinformatics

1. What is the main advantage of using signatures identified from the single cell RNA sequencing, comparing to the previous study by Doebley et al. published the same journal, titled “A framework for clinical cancer subtyping from nucleosome profiling of cell-free DNA”?

We thank Reviewer #3 for referring to Doebley *et al*'s work in the context of ours. We are aware of this study and agree that it is relevant to ours; in particular, because it is one of the only other WGS method to our knowledge that does not rely on tumor-associated copy number changes and is demonstrated to perform well at ultra-low sequencing depths for early-stage cancer classification. With that, the method from Doebley *et al* has several relevant differences from ours: 1. They calculate three fragment coverage metrics at the binding sites of 270 transcription factor to estimate gene regulatory activity, rather than assessing nucleosome periodicity in gene bodies. 2. They do not demonstrate that their features can be used for cell/tissue-of-origin analysis. Hence, they cannot pinpoint the cell types involved. 3. Doebley *et al* focus most of their analysis on tumor sub-typing in breast cancer by selecting differentially expressed transcription factors (TFs) between blood cells and breast cancer for downstream analysis. 4. Griffin also appears to be very sensitive to batch effects between datasets due to different sequencing workflows. This prevented them from being able to generalize their cancer prediction models across cohorts and required them to build new models for each dataset. We achieve a good performance when applying our colorectal cancer model in an external validation cohort demonstrating that our cell type ranks may be more stable across different sequencing platforms and depth.

To fully address Reviewer #3's comment and to respond to requests to benchmark our method against other WGS methods, we applied Griffin on our cancer cohorts for cancer classification. For this analysis, we cloned the git repository (https://github.com/adoebly/Griffin_analyses) and followed the Griffin wiki (<https://github.com/adoebly/Griffin/wiki>) for data processing. We used the 30,000-sites files for 270 transcription factors as is suggested by the authors for generic cancer classification. Griffin outputs 810 features per sample (3x270) which we used as a test dataset for their LUCAS-trained model which is made available to future users. Griffin

performed better than our cell type features in the context of high-coverage later-stage colorectal cancer (AUROC 90.9% compared to 84.7%) but worse for high-coverage early-stage breast cancer (AUROC 77.4% compared to 90.2%) and for low-coverage later-stage multiple myeloma (AUROC 75.6% compared to 95.0%). We have added ROC curves for the Griffin model to Figure S10 and an explanation to the main results section lines 301 – 308 and the methods section lines 571 – 578.

2. For classification task based on support vector machine, the input features comprising 456 cell types to classify cancer. However, the sample size (n = 16 and 52, for colorectal cancer and breast cancer) seems very limited relative to the number of features. Hence, the actual performance might be still unanswered. It will be informative to use unsupervised methods such as PCA or tSNE to visualize the data structure and check whether the patients will be clustered together depending on the pathophysiologically states using these features.

We thank Reviewer #3 for their suggestion and agree that it is informative to use unsupervised methods. We have performed an additional set of exploratory analyses using unsupervised methods. Principal component analysis (PCA) was performed, followed by Walktrap community detection to define clusters with fixed parameters (the nearest number of nodes was 3 with a walk step of 2). To visualize the dataset in lower dimensions, t-distributed stochastic neighbor embedding (tSNE) was used. We performed PCA with clustering and tSNE visualization for each disease cohort + matched controls using all 456 and 498 features as input for the cancer and preeclampsia cohorts, respectively.

We included these results in Figure S7 and the main text lines 269 - 274. We are encouraged to see that even in the unsupervised space there is separation between cases and controls. There is an uneven distribution of cases and controls in the walktrap community clusters and apparent separation between cases and controls in the tSNE visualization. In particular, pre-eclampsia cases are largely present in cluster 1, multiple myeloma cases in clusters 4 & 6, breast cancer cases in clusters 2 & 5, and colorectal cases in clusters 1, 12, & 16. We would also like to note that we chose to use the support vector machine for supervised learning because it is known to deal well with high dimensional data with fewer samples better than other machine learning algorithms.

3. It will be interesting to explore what is the minimal number of cell types required for classification of cancer with a desired performance. From this study, it is not clear to the reviewer as to how much improvement in window protection score based cfDNA nucleosome spacing analysis is contributed by the use of single cell RNA sequencing, if compared with the traditional bulk RNA sequencing.

To address Reviewer #3's comment, we have performed benchmarking against bulk tissue RNA sequencing reference datasets. For the bulk tissue transcriptomic references, we downloaded the

consensus transcript expression levels summarized per gene in 50 tissues based on transcriptomics data from Human Protein Atlas (HPA) and GTEx (https://www.proteinatlas.org/download/rna_tissue_consensus.tsv.zip). These tissues largely correspond to the tissues from the Tabula Sapiens database for which we have single cell data and include placenta. In our cohorts, using bulk tissue features instead of cell type features decreases the AUROC from 95.0% to 87.1%, 84.7% to 73.4%, 90.2% to 84.6%, and 88.3% to 61.5% for multiple myeloma, colorectal cancer, breast cancer, and preeclampsia, respectively. We have added ROC curves for the bulk tissue models to Figure S8 and an explanation to the main results section lines 287 – 294 and the methods section lines 556 - 564.

4. In Fig. 2D, both uterus and prostate showed significant ranks, for example, above intestines. Did author compare these tissue contributions (uterus and prostate) between female and male subgraphs?

This may be an oversight from the reviewer. This is what we show in Figure 3B.

Reviewer #4 (Remarks to the Author): expertise in colorectal cancer genomics with single cell

Major comment

1. The authors acknowledged that this study builds on the work by Snyder et al (2016) by utilizing single cell transcriptome atlases instead of bulk tissue references. The method descriptions for WPS and FFT with respect to gene expression are relatively brief. Please elaborate on any differences from Snyder et al.'s methods, if present, clearly indicating what has been modified, why these modifications were made, and the resulting benefits.

We calculated window protection scores and performed nucleosome peak calling exactly as specified by Snyder et al (2016) using source code from the shendurelab GitHub (<https://github.com/shendurelab/cfDNA>). No changes were made to this workflow except that we used BAMs aligned to hg38 instead of hg19 and so a different reference genome was used. We have added this statement to the Methods section lines 491 - 495 and lines 510. For the gene expression analysis instead of using this file (<http://www.proteinatlas.org/download/rna.csv.zip>) used by Snyder *et al* we generated our own reference file based on gene expression values from the Tabula Sapiens database summarized per annotated cell type. We now have a section in the methods section dedicated to the preparation of this file (please see below).

2. Preparation of the Single cell transcriptome atlas should be described more in detail for the readers to repeat the process.

We would like to thank Reviewer #4 for their comment. We agree that more detailed information should have been provided on the preparation of this reference file for readers to repeat this process. We have added an additional section to Methods called “Preparation of the single cell

transcriptomic reference datasets” (lines 518 – 552) that provides download links to the publicly available data and step-by-step instructions for data processing.

3. Prediction sensitivity and specificity are known to differ depending on the study population. It would be good to use other cfDNA analysis (methylation etc) to compare performance.

We thank Reviewer #4 for the suggestions and agree that benchmarking is an important exercise. We have performed benchmarking against bulk tissue transcriptomic references (Snyder *et al*, 2016), ichorCNA (Meyerson *et al*, 2017), and Griffin (Doebler *et al*, 2022).

For the bulk tissue transcriptomic references, we downloaded the consensus transcript expression levels summarized per gene in 50 tissues based on transcriptomics data from Human Protein Atlas (HPA) and GTEx (https://www.proteinatlas.org/download/rna_tissue_consensus.tsv.zip). These tissues largely correspond to the tissues from the Tabula Sapiens database for which we have single cell data and include placenta. This is the updated version of the file that Snyder *et al* used in their analysis in 2016 (<http://www.proteinatlas.org/download/rna.csv.zip> -- no longer available). In our cohorts, using bulk tissue features instead of cell type features decreased the AUROC from 95.0% to 87.1%, 84.7% to 73.4%, 90.2% to 84.6%, and 88.3% to 61.5% for multiple myeloma, colorectal cancer, breast cancer, and preeclampsia, respectively. We have added ROC curves for the bulk tissue models to Figure S8 and an explanation to the main results section lines 287 – 294 and the methods section lines 556 - 564.

For the ichorCNA analysis we cloned the git repository (<https://github.com/broadinstitute/ichorCNA>) and followed the ichorCNA wiki (<https://github.com/broadinstitute/ichorCNA/wiki>) for data processing. We used the readCounter function from HMMCopy to generate wig files for all cases and controls. IchorCNA tumor fractions were then estimated using the runIchorCNA.R script for cases and controls using the default panel of normal reference. IchorCNA was able to distinguish between cases and controls with poorer performances compared to our cell type features with AUCs of 92.6 for multiple myeloma, 64.2% for colorectal cancer, and 65.7 for breast cancer. This highlights the major advantage of our method in patients with low tumor burden or copy number neutral cancers. We

have added ROC curves for ichorCNA to Figure S9 and an explanation to the main results section lines 296 – 301 and the methods section lines 566 - 571.

We have also performed benchmarking against fragmentation-based cfDNA method “Griffin” (Doelbey et al, 2022) as it does not rely on copy number alterations and has been demonstrated to perform well in early-stage cancers sequenced at low coverage. Griffin instead uses 3 fragment coverage metrics at the binding sites of 270 transcription factors to capture gene regulatory information. For this analysis, we cloned the git repository (https://github.com/adoebley/Griffin_analyses) and followed the Griffin wiki (<https://github.com/adoebley/Griffin/wiki>) for data processing. We used the 30000-sites files for 270 transcription factors as is suggested by the authors for generic cancer classification. Griffin output 810 features per sample (3x270) which we used as a test dataset for their model. Griffin performed better than our cell type features in the context of high-coverage later-stage colorectal cancer (AUROC 90.9% compared to 84.7%) but worse for high-coverage early-stage breast cancer (AUROC 77.4% compared to 90.2%) and for low-coverage later-stage multiple myeloma (AUROC 75.6% compared to 95.0%). Overall, we think Griffin is a useful framework, but has some limitations that our cell type features do not. A major difference is that Doelbey *et al* do not present a way to utilize their features for tissue-of-origin analysis and cannot be used to determine the location of tumors or involved cell types. We have added ROC curves for the Griffin model to Figure S10 and an explanation to the main results section lines 301 – 308 and the methods section lines 571 - 578.

While we think it is fair to compare against WGS methods, further requests to benchmark against methylation features are well beyond the scope of this study. Currently, methylation-based methods require workflows that are not readily transferrable to most clinical settings, whereas low-pass whole-genome cfDNA sequencing data is routinely and cheaply generated in some clinical settings, particularly prenatally (80% of pregnant women in Belgium/Netherlands).

4. Specific comments for the figures

Fig 2G shows the down sampling results for the highest and lowest. What about other cell types? What about cell types used for the classification experiments?

We thank Reviewer #4 for requesting information on other cell types. We have only included downsampling results for the highest and lowest correlations in the main figure panel (2E) for ease of visualization, but we agree that it is useful to include information on all 456 cell types. We now added Figure S3 (copied below) which includes information on the rank of all 456 cell types per down sampling level. We correlated cell type ranks at the highest down sampling level (40%) with the ranks for those cell types at every other down sampling level. This analysis demonstrates that cell type ranks become less stable at lower read depths, but largely retain their rank placement down to a 0.1% down sampling of reads (4M, 0.1-fold coverage). Below this cutoff the strength of correlation between cell type ranks and their original rank drops.

Fig 3C : which specific cell types within placenta?

We have added information on the placental dataset to the Methods section in the “Preparation of the single cell transcriptomic reference datasets” (lines 536 – 538). These are placental cell types captured and annotated by Vento Tormo *et al* (2018) from the placental compartment of the maternal-fetal interface: syncytiotrophoblast, villous cytotrophoblasts, proliferative villous cytotrophoblasts, extravillous cytotrophoblasts, proliferative extravillous cytotrophoblasts, Hoffbauer cells, and two separate fetal fibroblast populations.

Fig 3D : what about the rank of other cell types?

We performed the same correlation analysis for 498 cells types as described above for “GC02” and included it as Figure S4. Please refer above for an explanation of the additional down sampling figures.

Fig 3F: Statistics?

We thank Review #4 for catching that. We have added standard deviation error bars to plot 3F, and report the Kruskal Wallis, p-value 0.023 in the results section line 210.

Fig 4F: Stage of multiple myeloma?

We thank Reviewer #4 for noticing that that information was missing. Our multiple myeloma cohort was 25% stage I, 54% stage II, and 21% stage III. We have added the multiple myeloma stages to the text lines 253. Patients were staged using the Revised Multiple Myeloma International Staging System (R-ISS)⁵.

Fig 4B,D,F: what are the ranges of healthy controls?

The range of healthy control ranks for these cell types is significantly lower than those of the cases as demonstrated in case-control analysis visualized in the Volcano plot. Foldchange and p-values for case-control comparison for these cell types are included in Tables S1, S2, and S3. For ease of visualization we have not included data points for control values in this plot as we feel the information is already presented in Figures 4 panels A, C, and E and Tables S1, S2, and S3. If the reviewer meant ichorCNA ranges for the healthy controls, we have now calculated the ichorCNA values for the controls and included them in Table S5. These ichorCNA values were used to distinguish cases from controls in a benchmarking exercise that yielded an AUROC of 92.6%, 65.7%, and 64.2%, for multiple myeloma, breast cancer, and colorectal cancer, respectively.

References

1. De Borre, Marie et al. “Cell-free DNA methylome analysis for early preeclampsia prediction.” *Nature medicine* vol. 29,9 (2023): 2206-2215. doi:10.1038/s41591-023-02510-5
2. Chan, Rebecca W Y et al. “Plasma DNA aberrations in systemic lupus erythematosus revealed by genomic and methylomic sequencing.” *Proceedings of the National Academy of Sciences of the United States of America* vol. 111,49 (2014): E5302-11. doi:10.1073/pnas.1421126111
3. Vogelstein, B. *et al.* The origin of highly elevated cell-free DNA in healthy individuals and patients with pancreatic, colorectal, lung, or ovarian cancer. *Cancer Discov.* 13,10 (2023).
4. Dor, Y. *et al.* Remote immune processes revealed by immune-derived circulating cell-free DNA. *eLife.* 10:e70520 (2021).
5. Palumbo, Antonio et al. “Revised International Staging System for Multiple Myeloma: A Report From International Myeloma Working Group.” *Journal of clinical oncology : official journal of the American Society of Clinical Oncology* vol. 33,26 (2015): 2863-9. doi:10.1200/JCO.2015.61.2267

REVIEWERS' COMMENTS

Reviewer #1 (Remarks to the Author):

Thank you for your detailed responses to mine and other Reviewers comments. I have gone through these and think that you have addressed my comments and queries adequately. I think that the amendments have improved my understanding on the work that you are reporting on, and the potential future clinical applications for the technology. I would agree that the amendments have improved the paper overall. I have no further comments from my perspective.

Reviewer #2 (Remarks to the Author):

I thank the authors for their detailed analysis and responses to the comments and questions. These additional analyses have improved the paper and answered all my questions, and I congratulate the authors on their work.

Reviewer #3 (Remarks to the Author):

The authors have adequately addressed the concerns.

Reviewer #4 (Remarks to the Author):

The authors thoroughly addressed my comments.